# Inverse Depth Scaling From Most Layers Being Similar

Yizhou Liu [1]    Sara Kangaslahti [2]    Ziming Liu [1 3]    Jeff Gore [1]

## Abstract

Neural scaling laws relate loss to model size in large language models (LLMs), yet depth and width may contribute to performance differently, requiring more detailed studies. Here, we quantify how depth affects loss via analysis of LLMs and toy residual networks. We find loss scales inversely proportional to depth in LLMs, probably due to functionally similar layers reducing error through ensemble averaging rather than compositional learning or discretizing smooth dynamics. This regime is inefficient yet robust and may arise from the architectural bias of residual networks and target functions incompatible with smooth dynamics. The findings suggest that improving LLM efficiency may require architectural innovations to encourage compositional use of depth.[1]

## 1. Introduction

Neural scaling laws played an important role in the success of today's large language models (LLMs), revealing how model performance predictably improves with increases in model size and dataset size. Quantitatively, loss is found to have power-law relationships with these factors (Hestness et al., 2017; Kaplan et al., 2020; Hoffmann et al., 2022). Yet, the origin of these power laws remains largely unclear.

Studies on neural scaling laws have primarily focused on total parameter count, treating model size as a monolithic quantity (Spigler et al., 2020; Bordelon et al., 2020; Hutter, 2021; Maloney et al., 2022; Sharma & Kaplan, 2022; Michaud et al., 2023; Liu et al., 2025b; Bordelon et al., 2025a). In practice, there is another line of research studying the optimal relationship between model width and depth (Safran & Shamir, 2017; Levine et al., 2020), indicating their different contributions to loss. Width heuristically

limits the representation capacity, while depth restricts the quality of transformation. Recently, theoretical works on neural scaling laws also suggested separating the scaling with model size into one with width and another with depth (Liu et al., 2025a; Bordelon et al., 2025b). In this work, we try to dive into the scaling with depth, asking:

> **Q: Neural depth scaling?**
>
> How do LLMs use their depth, and what is the quantitative law between depth and performance?

There have been many empirical investigations on how LLMs utilize their layers (Lad et al., 2024; Lioubashevski et al., 2024; Gromov et al., 2024; Sanyal et al., 2024; Men et al., 2025; Sun et al., 2025; Csordás et al., 2025; Gupta et al., 2025; Hu et al., 2025). Despite the differences in approaches and analysis, these works conclude that LLMs use their depth inefficiently: a lot of layers are redundant or make incremental updates on hidden states. However, while these studies provide compelling evidence of depth inefficiency, they lack a quantitative framework for understanding why this inefficiency exists and what scaling relationship depth follows.

Different theories about how depth can contribute to neural network performance have been proposed. One of the fundamental intuitions for multiple layers in neural networks is that depth allows learning hierarchical abstractions through composition (Bengio et al., 2009; 2013; LeCun et al., 2015). For LLMs, this would correspond to layers building increasingly abstract representations, with early layers handling syntax, middle layers capturing semantics, and deeper layers performing complex reasoning (Levine et al., 2020; Cagnetta et al., 2024; Csordás et al., 2025). The loss scaling in this regime may then heavily depend on the data properties. We will call this perspective on depth usage **compositional assembly** (Figure 1).

A second perspective views deep residual networks as discrete approximations of smooth dynamical systems, i.e., neural ODEs (Chen et al., 2018; Sander et al., 2022; Chizat, 2025), which we refer to as **procedural assembly**. In this regime, layers perform incremental refinement along a smooth path, and the loss arising from discrete approximations is often a power law with depth or number of

[1]Massachusetts Institute of Technology [2]Harvard University [3]Stanford University. Correspondence to: Yizhou Liu <liuyz@mit.edu>, Jeff Gore <gore@mit.edu>.

*Proceedings of the 43rd International Conference on Machine Learning*, Seoul, South Korea. PMLR 306, 2026. Copyright 2026 by the author(s).

[1]Code available at github.com/liuyz0/DepthScaling

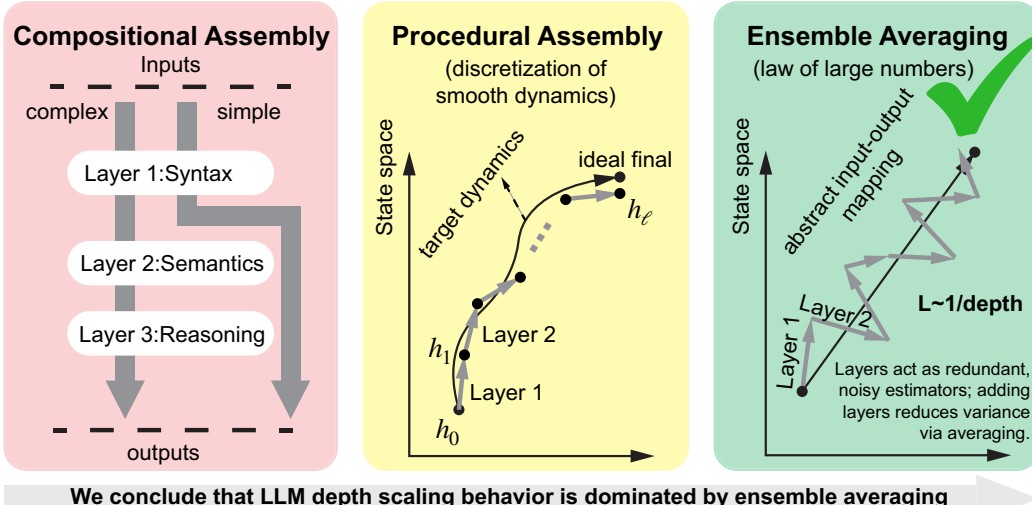

*Figure 1.* Three regimes of how LLMs may utilize their layers.

integration "time" grid points.

Finally, a third possibility is that layers function as an ensemble of similar shallow subnetworks (Veit et al., 2016; Lu et al., 2020), each contributing to the outcome with the same mean but a different error that can be partly canceled through averaging. In this regime of depth utilization, which we call **ensemble averaging**, the loss will be a power law with depth due to the central limit theorem (Bahri et al., 2024; Song et al., 2024).

Although there has been empirical understanding of how LLMs utilize their depth and theoretical expectations on loss scaling with depth, the connection between LLM reality and theory has not yet been established. To have a quantitative framework of the actual depth scaling of LLMs, we evaluate the behaviors of hidden states across layers in LLMs and compare with theory (Section 2). We find from hidden states that most layers in LLMs are not in the compositional assembly regime. Combining such insights with previous works, we then propose a power-law loss decomposition and find that the depth-limited part is roughly inversely proportional to depth. To understand the inverse depth scaling, we design toy model training experiments (Section 3) to study both the loss and hidden states in the procedural assembly and the ensemble averaging regimes (Section 4). After comparing our controlled experiments with LLM evaluations, we conclude that LLMs utilize most layers similarly to the ensemble averaging regime. We compare our findings to previous works in Section 5, and discuss limitations and implications in Section 6.

To summarize, our work provides a more quantitative explanation for how LLMs utilize their layers and pushes neural scaling laws to a more detailed level. Our findings have

direct implications for the optimal depth-width relationship and future directions of architectural innovations.

> **Our contributions are:**
>
> • We find the inverse depth scaling of loss in LLMs.
> • We find a mechanistic reason for this scaling behavior: similar layers form an ensemble, reducing error by averaging.

## 2. LLM Experiments

To connect theoretical expectations with the behavior of real LLMs, we design experiments to probe how hidden states evolve across layers. Among the three regimes, compositional assembly should exhibit qualitatively distinct signatures. If different layers encode progressively higher levels of abstraction, simpler inputs should require fewer layers: some hidden states may converge or stop changing early, while others continue to evolve deeper into the network, depending on the input complexity. In contrast, procedural assembly and ensemble averaging are unlikely to show sharp depth-wise stopping. Procedural assembly approximates smooth dynamics, while ensemble averaging implies many layers perform similar transformations.

Under compositional assembly, networks of different depths should share similar early-layer representations at matched layer indices, since these layers encode the same lower-level features; deeper networks would then add new layers that capture higher-level structures inaccessible to shallower models. By contrast, in procedural assembly or ensemble averaging, the per-layer update magnitude should systematically decrease as depth increases, making more fine-grained

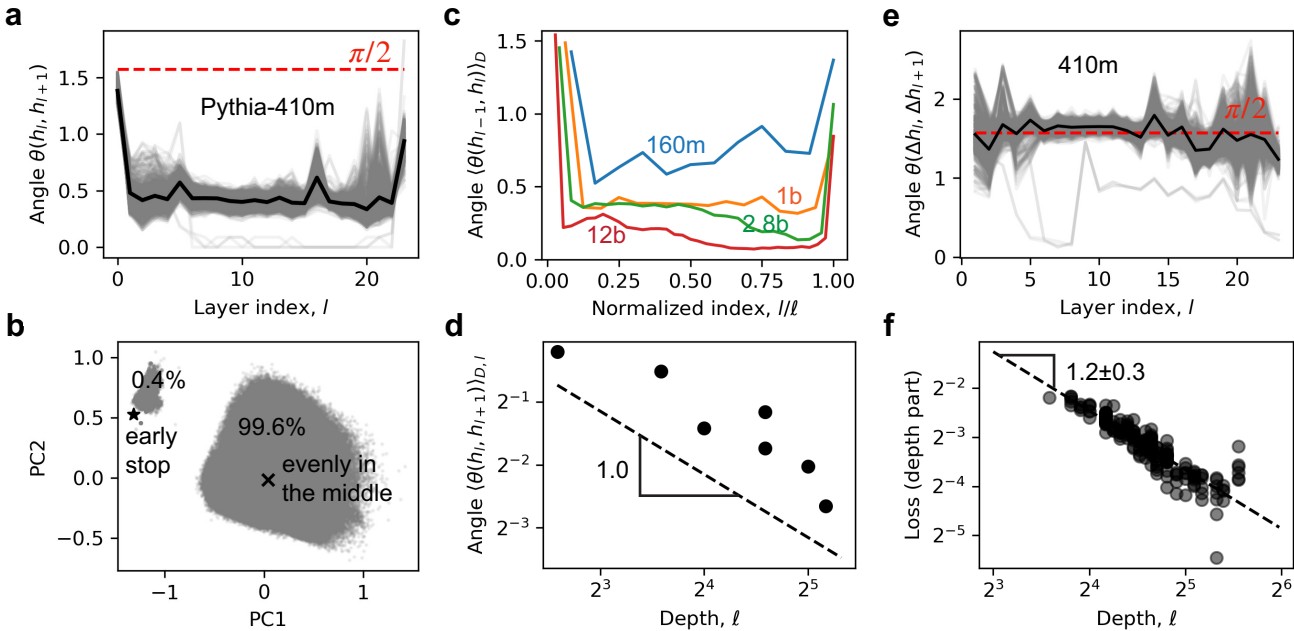

*Figure 2.* Most data in most layers are processed in an even and incremental way. (a) Updates of hidden states measured by the angle between neighboring hidden states $\theta(h_l, h_{l+1})$ show incremental changes in most middle layers. (b) By PCA of $\theta(h_l, h_{l+1})$, most tokens are updated evenly in the middle, a small fraction stops updating early, which usually corresponds to the first tokens in documents. (c) Mean update decreases with depth. (d) The mean updates scales approximately inversely proportional to depth, suggesting more fine-grained updates rather than decomposing higher-level information. (e) Correlation between neighboring updates is small, suggesting non-smooth dynamics. (f) LLM loss roughly follows an inverse depth scaling. More details about hidden state analysis are in Section A, and those of loss fitting are in Section B.

and accurate estimations, rather than introducing qualitatively new representations. **Our experiments thus focus on the hidden state updates across layers for each input.**

We start by studying the angular change of hidden states across layers for individual data points. Due to the wide use of layer norms in LLMs (Brown et al., 2020; Biderman et al., 2023; Zhang et al., 2022; Yang et al., 2024; Jordan et al., 2024), the angular updates of hidden states capture the essential dynamics. Given an input token $x$ at a certain position in the document, the embedding matrix first turns the token into a vector $h_0$, Transformer layer $l$ then update $h_{l-1}$ (at the same position along context length direction) into $h_l$ for $l = 1, 2, ...\ell$ where we use $\ell$ to represent the depth or the number of layers, and at the end, $h_\ell$ will go through the final layer norm and the language model head to generate logits for next-token prediction. In each Transformer layer, e.g., layer $l$, information about previous tokens can be grabbed into $h_l$. Yet, we still refer to $h_l$ as the hidden state of the input token $x$ after layer $l$. We calculate the angle between neighboring hidden states, denoted as $\theta(h_l, h_{l+1})$ for analysis (details in Section A). If the hidden states are following smooth dynamics in the depth direction, $\theta(h_l, h_{l+1})$ is related to the first-order derivative.

We show the angles between hidden states, $\theta(h_l, h_{l+1})$,

measured from Pythia-410m (Biderman et al., 2023) and evaluated on the FineWeb dataset (Penedo et al., 2024) in Figure 2a. Each gray line represents $\theta(h_l, h_{l+1})$ as a function of $l$ for a single input token, while the dark line shows the dataset-averaged angle, denoted as $\langle\theta(h_l, h_{l+1})\rangle_\mathcal{D}$. We observe that only a few inputs exhibit early stopping of hidden-state updates, reminiscent of compositional assembly. The first and last layers rotate hidden states by large angles close to $\pi/2$, in contrast to intermediate layers. Most middle layers update hidden states by similarly small angles for the majority of inputs, resembling procedural assembly and ensemble averaging. The norms of hidden states convey similar messages: the first and last layer changes norm a lot, and the middle layers do not update the norms much (Section A). Overall, these observations suggest that **no single theoretical regime fully explains LLM behavior; instead, one regime may dominate while others coexist.**

We proceed to quantify the ratios of different behaviors in Pythia-410m. A trajectory $\theta(h_l, h_{l+1})$ with $l$ from 0 to $\ell - 1$ can be thought of as an $\ell$-dimensional vector. And there are $|\mathcal{D}|$ (the number of evaluated tokens) such vectors. We perform principal component analysis (PCA) of these vectors (Section A) and find that there are two clusters (Figure 2b). To interpret the two clusters, we construct two ideal trajectories. One trajectory is called "early stop", whose

angles in the middle are zero, and another one is "evenly in the middle" with angles in the middle being $0.45$ rad. The "early stop" trajectory is projected into the subspace of the first two components, denoted by the star, and the "evenly in the middle" one is the cross (Figure 2b). We can then conclude that the small cluster, which accounts for $0.4\%$ of the tokens, roughly contains data not processed in the middle, and the big cluster, which occupies the remaining $99.6\%$, contains data that is approximately evenly updated in the middle layers. We check some of the tokens whose trajectories are in the small cluster and find that they are the first tokens of the documents. These behaviors are generally true for other model sizes and model families (Section A). Since we study the loss averaged over all data, and how it scales with depth, **we can focus on the majority of tokens – those being "evenly updated in the middle".**

We average the angle updates across tokens to identify the dominant depth-wise behavior. The dataset-averaged angles $\langle\theta(h_{l-1}, h_l)\rangle_{\mathcal{D}}$, for $l = 1, 2, \ldots, \ell$, are plotted against the normalized layer index $l/\ell$ in Figure 2c for different Pythia models. We find that the first and last layers exhibit similar large update angles across model depths, whereas the intermediate layers show systematically decreasing angles as depth increases. This suggests that the first and last layers may implement depth-independent functions, reminiscent of compositional assembly. However, since our focus is on depth scaling, the dominant contribution comes from the bulk of the layers. For these middle layers, increasing depth corresponds to finer-grained or more accurate incremental updates, consistent with procedural assembly or ensemble averaging. We average $\langle\theta(h_{l-1}, h_l)\rangle_{\mathcal{D}}$ over the middle layers $l = 2, 3, \ldots, \ell-1$, denoted as $\langle\theta(h_l, h_{l+1})\rangle_{\mathcal{D},l}$, and find that this quantity scales approximately inversely with depth (Figure 2d). This scaling further supports the procedural assembly or ensemble averaging interpretation. We therefore conclude the following

---

**Result 1: Not compositional assembly**

For the majority of data and the majority of layers, LLM depth behavior may be best described by procedural assembly or ensemble averaging.

---

To distinguish procedural assembly from ensemble averaging in the middle layers, we evaluate the correlation of incremental hidden-state updates across layers. We denote the update produced by Transformer layer $l$ as $\Delta h_l = h_l - h_{l-1}$, and quantify inter-layer correlation by the angle between neighboring updates, $\theta(\Delta h_l, \Delta h_{l+1})$. We find that correlations are generally weak: $\theta(\Delta h_l, \Delta h_{l+1})$ is typically large, which is inconsistent with smooth dynamics (Figure 2e). However, this conclusion is not definitive, as we lack a calibrated reference for comparison. Controlled training experiments with known data-generating processes are re-

quired to disentangle procedural assembly and ensemble averaging (later Sections), after which detailed signatures can be compared to LLM measurements to draw a definitive conclusion.

In addition to hidden-state dynamics, we can directly examine loss scaling in LLMs. Neural scaling laws are largely empirical and are typically described by the following form in previous works (Hoffmann et al., 2022):

$$L = \frac{c_N}{N^{\alpha_N}} + \frac{c_D}{D^{\alpha_D}} + L_0, \tag{1}$$

where $N$ is the number of parameters, $D$ is the training dataset size, and $L_0$ is the irreducible loss associated with the entropy of the target distribution. Here, following our findings and prior theoretical arguments (Levine et al., 2020; Liu et al., 2025a; Bordelon et al., 2025b), we suggest that the model-size term should be decomposed into width- and depth-dependent contributions:

$$\frac{c_N}{N^{\alpha_N}} = \frac{c_m}{m^{\alpha_m}} + L_\ell + \cdots, \tag{2}$$

where $m$ denotes the model width, which leads to a power-law loss term under multiple theories (Bahri et al., 2024; Liu et al., 2025a; Bordelon et al., 2025b), $L_\ell$ is a term that depends only on depth, and the ellipsis denotes additional cross terms involving both width and depth. Conceptually, the width-dependent power law captures representation-limited error that cannot be eliminated even with perfect learning of transformations, while $L_\ell$ captures transformation-limited error that cannot be eliminated by perfect feature representation. Cross terms arise from the interplay between imperfect representation and imperfect transformation, and are expected to be higher order and subdominant when both $m$ and $\ell$ are large. We therefore focus on the dominant contributions. Some theories predict that $L_\ell$ decays exponentially with depth, and that the optimal width-depth trade-off under a fixed parameter budget $N \propto m^2\ell$ implies depth scaling logarithmically with width (Levine et al., 2020). Other theories instead predict a power-law dependence on depth. We argue that if, near the optimal width-depth relation, the loss exhibits a power-law dependence on $N$ without logarithmic corrections, then $L_\ell$ must itself scale as a power law in $\ell$. Motivated by prior experiments and theory, we propose the following decomposed scaling form:

$$L = \frac{c_m}{m^{\alpha_m}} + \frac{c_\ell}{\ell^{\alpha_\ell}} + \frac{c_D}{D^{\alpha_D}} + L_0. \tag{3}$$

We next fit real LLM data using this decomposed scaling form. The model contains seven free parameters, but most open-source model families do not provide sufficient coverage in both width and depth to reliably constrain all terms. Moreover, differences across model families introduce confounding architectural and training variations, complicating

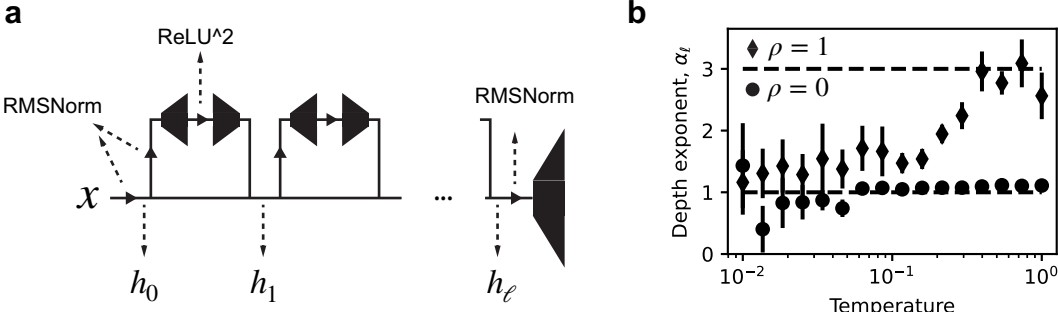

*Figure 3.* The toy model can exhibit inverse depth scaling when the underlying transformation dynamics to learn are smooth, and the target distribution is sharp, or when the underlying dynamics to learn are noisy. (a) Architecture of the toy model. (b) Tied teacher weights ($\rho = 1$) produce smooth dynamics and yield $\alpha_\ell = 1$ at low teacher temperature (peaked output distributions). Independent teacher weights ($\rho = 0$) generate non-smooth dynamics and have $\alpha_\ell = 1$ across different temperatures. Error bars are standard errors. Details are in Section C.

joint fitting. We therefore use around 200 data points reconstructed from the Chinchilla models (Hoffmann et al., 2022; Besiroglu et al., 2024) for our fitting. We fit the model by minimizing the mean squared error of the logarithm of the loss and obtain $\alpha_m = 0.98 \pm 0.08$, $\alpha_\ell = \mathbf{1.2 \pm 0.3}$, and $\alpha_D = 0.30 \pm 0.01$ (see Section B). The relative error between the empirical loss and the fitted model prediction is $0.4\%$ on average, indicating a good fit and providing partial support for the proposed decomposition. The depth-dependent loss component, defined as the residual after subtracting the fitted irreducible, width-dependent, and dataset-size-dependent terms, is shown in Figure 2f and exhibits approximately power-law dependence on depth. The fitted width exponent $\alpha_m \approx 1$ is consistent with a recent observation (Liu et al., 2025a), and $\alpha_D = 0.30$ closely matches the original Chinchilla scaling exponent (Hoffmann et al., 2022). We therefore conclude that

> **Result 2: Empirical inverse depth scaling**
>
> Equation (3) provides a reasonable description of LLM loss, with the depth-dominant contribution scaling approximately inversely with depth.

Both the procedural assembly and the ensemble averaging regimes can lead to power-law scaling. A more detailed understanding is needed to distinguish them.

## 3. Toy Model

To study the procedural assembly and ensemble averaging regimes in isolation, we design controlled experiments on toy models and complement them with theoretical analysis. Theory alone cannot determine which mechanisms are relevant in realistic training, while experiments on full-scale LLMs are prohibitively expensive and confounded by many interacting factors. Toy models then provide a controlled

setting in which specific hypotheses can be tested directly.

Our toy model captures a key architectural ingredient of Transformers: residual connections. We adopt a teacher-student setup, in which the teacher generates training data for the student. The teacher and student share the same architecture, except that the teacher has greater depth. Inputs $x \in \mathbb{R}^m$ are sampled i.i.d. from the standard normal distribution. For concreteness, we describe the student architecture. The input $x$ is first normalized using root mean square norm to obtain the initial hidden state $h_0$ (Figure 3a),

$$h_0 = \text{RMSNorm}(x). \tag{4}$$

The hidden state then propagates through $\ell$ residual blocks,

$$h_l = h_{l-1} + \text{MLP}_l(h_{l-1}), \quad l = 1, \dots, \ell, \tag{5}$$

where $\ell$ is the student depth. Each residual block consists of a two-layer MLP,

$$\text{MLP}_l(v) = B_l \text{ReLU}^2(A_l \text{RMSNorm}(v) + b_l), \tag{6}$$

with parameters $B_l \in \mathbb{R}^{m \times 4m}$, $A_l \in \mathbb{R}^{4m \times m}$, and $b_l \in \mathbb{R}^{4m}$. The logits are obtained by projecting the final hidden state,

$$y = W \text{RMSNorm}(h_\ell) \in \mathbb{R}^n, \tag{7}$$

where $W \in \mathbb{R}^{n \times m}$. The teacher follows the same architecture but has depth $\ell^* > \ell$, producing logits $y^*$. We use an extra $*$ to denote the variables in the teacher. From these logits, we define output distributions and train the student by minimizing the KL-divergence[2] between the student and teacher distributions. We adopt these architectural designs from recent practices of LLMs (Jordan et al., 2024).

The toy model is a highly simplified abstraction of LLMs, but it is sufficient for isolating depth-scaling phenomena.

---

[2]KL-divergence is cross-entropy subtracting the entropy of target distributions, which is constant, and has no difference in scaling behaviors.

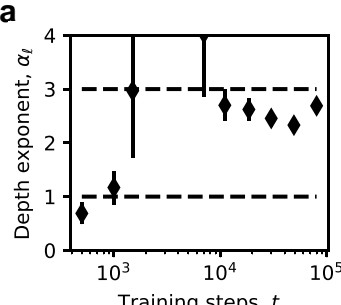
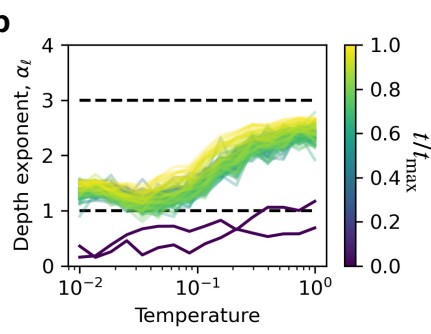
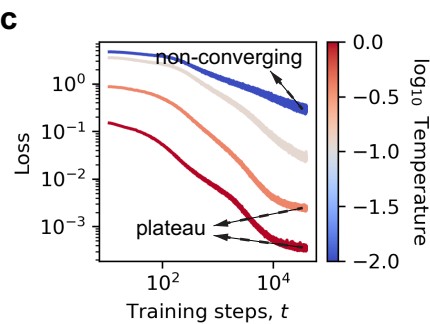

*Figure 4.* Matching smooth dynamics leads to the inverse depth scaling when not trained well. (a) The depth scaling exponent $\alpha_\ell$ is near 1 in the early stage of training but increases to 3 after training. Error bars are standard errors. (b) Low temperature makes training slow, yet all $\alpha_\ell$ tend to increase with more training. Maximum number of training steps $t_{\max} = 80000$. (c) Loss versus training steps for $\ell = 6$ from experiments in Figure 3 suggest that low teacher temperature requires longer training, and the corresponding student has not yet converged. Details in Section C.

We ignore embedding training and focus on transformation error, interpreting the toy-model input $x$ as the output of an embedding layer in LLMs. By using the same width $m$ for the teacher and student, we eliminate representation-limited error by construction, so that any residual error is transformation-limited. We also omit attention, which in real LLMs induces spatially coupled dynamics that are better described by PDEs rather than ODEs. Nevertheless, depth scaling alone should remain the same for "PDEs" and "ODEs". Other architectural components are kept as close as possible to standard LLM designs.

We tune teacher and data properties to induce different depth-scaling regimes. We fix $m = 32$, $n = 128$, teacher depth $\ell^* = 128$, and vary the student depth $\ell$ from 6 to 48. Teacher MLP weights are initialized with standard schemes and rescaled by $1/\sqrt{\ell}$ to ensure that the cumulative transformation from $h_0^*$ to $h_{\ell^*}^*$ remains $O(1)$. Among the controllable data properties, we find that the teacher "temperature" strongly affects training dynamics. After obtaining teacher logits $y^*$, we divide them by a temperature parameter before applying softmax to obtain the target distribution. Lower temperature increases the logit norm and produces sharper, more peaked target distributions. Another key control is whether teacher MLP weights are tied across layers, i.e., $A_l^*$ are identical, and $B_l^*$ are identical for all $l$. Weight tying induces smooth hidden-state dynamics in the limit $\ell^* \to \infty$, with derivatives of arbitrary order, and is expected to produce procedural assembly. In contrast, sampling teacher MLP weights i.i.d. across layers yields hidden-state dynamics resembling a random walk, which is non-smooth and likely leads to ensemble averaging for students. Students are trained for 40000 steps using Adam (Section A). We fit the loss using

$$L = \frac{c_\ell}{\ell^{\alpha_\ell}} + L_{\setminus \ell}, \qquad (8)$$

where $L_{\setminus \ell}$ denotes depth-independent contributions. For independent teacher weights (weight correlation $\rho = 0$ in Figure 3b), we observe inverse-depth scaling with $\alpha_\ell \approx 1$. For tied teacher weights ($\rho = 1$ in Figure 3b), the fitted depth exponent $\alpha_\ell$ increases from 1 to 3 as teacher temperature increases. These results indicate that both procedural assembly and ensemble averaging may yield inverse-depth scaling in certain regimes. We next analyze the mathematical origins of these scaling behaviors.

## 4. Toy Model Analysis

We first analyze the case where teacher weights are tied. In this setting, the underlying transformation can be modeled as a smooth continuous dynamics, and the student may implement procedural assembly, which we aim to test explicitly. The normalized layer index $s = l/\ell$ plays the role of "time" in a neural ODE, with step size $\Delta s = 1/\ell$. The residual dynamics for a given input can be written as

$$h(s + \Delta s) = h(s) + f(s)\Delta s. \qquad (9)$$

Assuming the teacher hidden states are smooth in $s$ (with $\ell^* \gg \ell$), we study the deviation between student and teacher trajectories:

$$h\left(\frac{l+1}{\ell}\right) - h^*\left(\frac{l+1}{\ell}\right) = h\left(\frac{l}{\ell}\right) - h^*\left(\frac{l}{\ell}\right)$$
$$+ \frac{1}{\ell}\left(f\left(\frac{l}{\ell}\right) - f^\circ\left(\frac{l}{\ell}\right)\right)$$
$$+ \int_{l/\ell}^{(l+1)/\ell}\left(f^\circ\left(\frac{l}{\ell}\right) - f^*(s)\right)\mathrm{d}s. \qquad (10)$$

Here, $f$ denotes the student transformation, $f^\circ$ is the optimal student transformation attainable by adjusting weights, and $f^*$ is the true teacher transformation. The second term in Equation (10) captures the error due to imperfect training to match the local transformation, while the third term

**a** 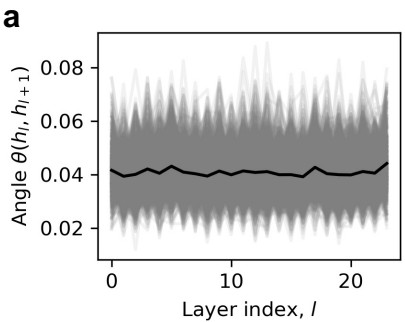

**b** 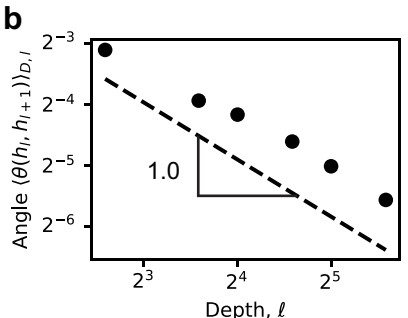

**c** 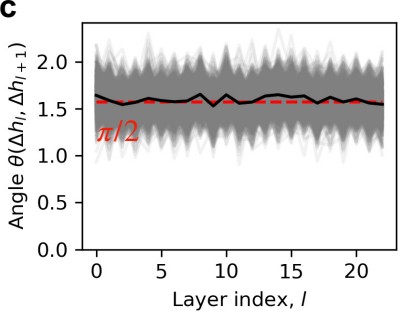

*Figure 5.* Matching the full transformation as an ensemble may explain the inverse depth scaling in LLMs. (a) Hidden state updates of a student whose teacher has independent MLP weights are even across layers. Each gray line is from one input, and the dark line is averaged on dataset. (b) Mean update in one layer scales inversely proportional to depth. (c) Correlation between neighboring updates is small, suggesting no smooth dynamics. These hidden state features agree with ensemble averaging and are similar to those in LLMs. Details in Section C.

corresponds to the discretization error. We next analyze the scaling behavior of these contributions.

If the error in matching the local transformation dominates, each student layer contributes an $O(1/\ell)$ error as indicated in Equation (10). With $\ell$ layers, naive accumulation yields $\|h(1) - h^*(1)\| = O(1)$. However, if errors across layers are independent, the central limit theorem implies diffusive accumulation, giving a **typical behavior**

$$\|h(1) - h^*(1)\| \overset{t}{=} O(1/\sqrt{\ell}). \tag{11}$$

When $h(1)$ is close to the optimal trajectory $h^*(1)$, we can Taylor expand the analytic loss around $h^*(1)$ where the gradient is zero, yielding that the leading loss contribution is quadratic in $\|h(1) - h^*(1)\|$. Consequently, the loss can range from an $O(1)$ upper bound in the worst case to a typical scaling $L \sim 1/\ell$.

Once training converges, the loss is dominated by discretization error. The residual stream corresponds to a first-order integration scheme, and for smooth $f^*$ we have $f^\circ(l/\ell) - f^*(s) = O(\Delta s)$ for $s \in [l/\ell, (l+1)/\ell]$. Consequently, the error contributed by a single student layer is $O(\Delta s^2)$, i.e., $O(1/\ell^2)$. In the worst case, errors accumulate coherently, yielding $\|h(1) - h^*(1)\| = O(1/\ell)$ and hence $L \sim 1/\ell^2$. Under independent layer-wise discretization errors, the typical behavior is $\|h(1) - h^*(1)\| \overset{t}{=} O(1/\ell^{3/2})$, which implies $L \sim 1/\ell^3$.

We test these theoretical predictions using experiments at a temperature 1, where training converges reliably (Section C). At early training times $t$, the fitted depth-scaling exponent is close to 1 (Figure 4a), consistent with the training-dominated behavior. During intermediate training, additional dynamics obscure a clean power-law fit in depth. At late training, however, the exponent approaches 3, in agreement with the predicted typical discretization-dominated scaling. We modify the student architecture to mimic a

higher-order integral scheme, which increases $\alpha_\ell$ as expected under tied teacher weights (Section C). These results confirm that tied teacher weights induce procedural assembly (discretization of smooth dynamics) in the student.

To understand why the scaling exponent reduces to 1 at low teacher temperature, we analyze $\alpha_\ell$ across temperatures and training steps (Figure 4b). We find that $\alpha_\ell$ increases with $t$ for all temperatures, but grows significantly more slowly at low temperatures. At high temperatures, the loss saturates by the end of training, whereas at low temperatures it continues to decrease, indicating incomplete optimization (Figure 4c). We thus attribute the observed $\alpha_\ell \approx 1$ regime to imperfect training: low temperature can map a wide range of hidden states to similar peaked output distributions, filtering information and reducing gradient signal magnitudes (Liu et al., 2026), thereby slowing convergence.

Since our primary focus is on depth scaling after training convergence, rather than cross terms involving training dynamics and depth (Equation (3)), we argue that procedural assembly is unlikely to be the dominant mechanism in LLM training, as it predicts a converged depth exponent $\alpha_\ell = 3$.

We next study the case of independent teacher weights. In this setting, $f^*(s)$ is non-smooth and its derivative does not exist. As a result, the discretization term in Equation (10) (third row) is $O(\sqrt{\Delta s})$ if $f^\circ$ attempts to match the local value $f^*(l/\ell)$. Even under typical error accumulation, this yields an $O(1)$ deviation in the final hidden state and a non-vanishing loss. Independent teacher weights therefore cannot yield procedural assembly. For general transformations that cannot be modeled as smooth dynamics, the network cannot decompose the target function into local incremental updates and may instead follow ensemble averaging:

$$h(1) - h^*(1) = \frac{1}{\ell} \sum_{l=0}^{\ell-1} \left( f(l/\ell) - \int_0^1 f^*(s) \mathrm{d}s \right). \tag{12}$$

In this picture, each layer (or subnetwork) attempts to approximate the full transformation with some error. Each layer contributes an $O(1/\ell)$ error, and summing over $\ell$ layers yields an upper bound $\|h(1) - h^*(1)\| = O(1)$, with a typical behavior $\|h(1) - h^*(1)\| \overset{t}{=} O(1/\sqrt{\ell})$ under independent layer-wise errors. Consequently, the typical loss scaling is $L \sim 1/\ell$, i.e., $\alpha_\ell \geq 0$ with $\alpha_\ell \overset{t}{=} 1$. Consistent with this prediction, we find that with independent teacher weights the measured $\alpha_\ell$ is robustly close to 1 (Figure 3b). Examining hidden states of trained students, we observe that layer-wise updates, quantified by $\theta(h_l, h_{l+1})$, are approximately uniform across depth (Figure 5a). The averaged update magnitude $\langle \theta(h_l, h_{l+1}) \rangle_{\mathcal{D},l}$ scales inversely with the number of layers (Figure 5b). Moreover, correlations between neighboring updates, measured by $\theta(\Delta h_l, \Delta h_{l+1})$, are large and close to $\pi/2$, rejecting the hypothesis of smooth dynamics (Figure 5c). Together, these hidden-state signatures and loss-scaling results indicate that independent teacher weights induce ensemble averaging.

The hidden state behaviors in Figure 5 are qualitatively similar to those of most data in most layers in LLMs (Figure 2). The quantitative prediction of inverse depth scaling from ensemble averaging also agrees with the LLM scaling result. Combining all the evidence, we conclude that

> **Result 3: Depth scaling from ensemble averaging**
>
> Inverse depth scaling in LLMs is likely due to ensemble averaging.

## 5. Related Works

Building on empirical studies of neural scaling laws (Hestness et al., 2017; Kaplan et al., 2020; Hoffmann et al., 2022), our work suggests a finer decomposition of scaling behavior into separate depth- and width-dependent contributions. Compared to recent theoretical works that propose such separations (Liu et al., 2025a; Bordelon et al., 2025b), we provide a direct measurement on LLMs.

Our conclusion that LLMs predominantly operate in the ensemble averaging regime is consistent with prior findings that Transformer layers are highly redundant or inefficient (Gromov et al., 2024; Sanyal et al., 2024; Sun et al., 2025; Men et al., 2025), fail to exploit compositional structures in data (Csordás et al., 2025), and are robust to layer deletion or permutation (Lad et al., 2024). We extend these observations by providing a quantitative description, i.e., the inverse depth loss scaling.

Previous theoretical analyses of residual networks (Chizat, 2025; Sander et al., 2022) primarily focus on worst-case error bounds. In contrast, our theory–experiment framework shows that the observed loss follows the typical regime,

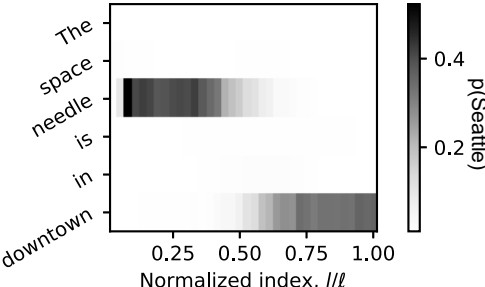

*Figure 6.* Factual statement prediction needs two functional groups. Causal tracing a single hidden state's effect on the correct prediction probability (Meng et al., 2022a) shows two important groups of layers having different functions. Within each group, ensemble averaging can happen. Results are obtained from Pythia-12b (details in Section D).

where layer-wise errors accumulate diffusively and are effectively independent.

## 6. Discussion

We find that loss scales roughly inversely proportional to depth in LLMs, and provide a mechanistic reason by a combination of experiments and theory: the majority of layers in LLMs make similarly incremental updates when facing most of the data and reduce error by averaging.

Our work is limited in several aspects. Loss decomposition Equation (3) was not derived from first principles but proposed based on a combination of previous empirical findings and theoretical insights, but is supported by the quality of fitting. The study of the interplay between width, depth, and training dataset size can be important, especially for small models.

We evaluated the three popular theories of layer utilization and concluded that ensemble averaging is more relevant via the hidden state behaviors and quantitative depth scaling. Due to the lack of first-principle derivations, we cannot rigorously exclude the possibility that there are other mechanisms that can produce the same inverse depth scaling and similar hidden state behaviors.

Our hidden state update analysis characterizes layer behavior statistically but cannot reveal the mechanistic function of individual layers. Although we reject compositional assembly, in which each additional layer handles a distinct level of abstraction, a weaker form of compositionality remains consistent with our findings: layers may organize into functional groups, with multiple layers sharing a similar role, and different groups serving different purposes. In the large-depth limit, the number of layers can far exceed the number of distinct functions required. Adding more layers then increases group sizes proportionally, and loss decreases

primarily through ensemble averaging within each group. This picture is consistent with our coarse-grained analysis and reinforces the conclusion that inverse depth scaling is driven mainly by averaging.

Mechanistic interpretability results support the functional-group picture and motivate further investigation in this direction. As an illustration, we reproduce a causal tracing experiment from Meng et al. (2022a) for factual memory localization (Figure 6; details in Section D). Given the prompt "The Space Needle is in downtown", the model should predict "Seattle". Corrupting the embeddings of "The Space Needle" causes the model to fail. We then restore a single hidden state from the uncorrupted run and measure the resulting probability of predicting "Seattle" as a function of hidden state location. Two distinct groups emerge: an earlier group retrieves related memories of "The Space Needle" by integrating the three tokens, while a later group extracts "Seattle" conditioned on "downtown". Within each group, the recovery probability fluctuates across layers, suggesting that the relevant knowledge is distributed rather than localized, which is consistent with noisy, redundant computation. Meng et al. (2022b) further showed that editing memories across a group of layers yields better outcomes than targeting a single layer. Ensemble averaging may therefore play an important functional role within each group.

We find the inverse depth scaling in LLMs and explain it by ensemble averaging, yet a more mechanistic explanation for the emergence of ensemble averaging is needed. The architectural bias of the residual connection may play an important role in not utilizing data compositionality (Veit et al., 2016). Next-token prediction may not be modeled by a smooth dynamical system, which forces the LLMs to use ensemble averaging.

The findings that fitted $\alpha_m \approx 1$ and $\alpha_\ell \approx 1$ imply that, under a fixed number of parameters $N \propto m^2\ell$, the optimal width-depth relationship is $m \propto \ell$. The result suggests that we should scale width and depth in a proportional manner, which is not rigorously true in practice (Section B), yet not far away. The inverse scaling has a direct implication on architectural choices. At the optimal width-depth ratio, both width scaling and depth scaling terms become proportional to $N^{-1/3}$. The predicted model size scaling exponent $1/3$ is close to the empirical $0.34$ reported in Chinchilla scaling (Hoffmann et al., 2022). The empirical scaling laws can be a consequence of the inverse depth scaling. Beyond current architectures, if we want to make LLMs more efficient in depth scaling, one direction is to encourage compositional use of depth. The use of recurrent depth seems to better explore data hierarchy (Geiping et al., 2025). We hope our insights can help the continued development of LLMs, from hyperparameter choosing to architecture design.

## Acknowledgements

The authors acknowledge the MIT Office of Research Computing and Data for providing high-performance computing resources that have contributed to the research results reported within this paper. Y. L. thanks Qiyao (Catherine) Liang and Yasaman Bahri for helpful discussions. S. K. is supported by the National Science Foundation Graduate Research Fellowship under Grant No. DGE 2140743. J. G. thanks the Schmidt Science Polymath Award for funding. The authors declare no competing interests.

## Impact Statement

This paper presents work whose goal is to advance the field of Machine Learning. There are many potential societal consequences of our work, none of which we feel must be specifically highlighted here.

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

# A. LLM Hidden States

## A.1. Methods

Our script is designed to probe how internal representations evolve across depth in a pretrained causal language model. The high-level goal is to take real text, run it through the model, and then compute layer-by-layer geometric and predictive diagnostics of the hidden states, such as norms, rotation angles, and the cross-entropy that would be obtained if we "stopped" at an intermediate layer and applied the model's final normalization and output head. The code runs entirely in inference mode (no training), and it accumulates per-token measurements over a large sample of documents before saving the resulting tensors for downstream analysis.

The experiment uses a streaming text source to avoid storing the full dataset locally. Specifically, it loads the `HuggingFaceFW/fineweb` dataset with `streaming=True` and repeatedly draws batches of raw text strings from the training split. The key data hyperparameters are `batch_size = 48` documents per step and `max_length = 1024` tokens per document (with padding and truncation enabled). The total number of processed documents is set as `num_docs = batch_size * 100 = 4800`, so the main loop runs for exactly `num_docs // batch_size = 100` iterations.

We can scan different models by changing the model name. The tokenizer is loaded from the same model name, and if no pad token is defined, the script assigns `tokenizer.pad_token = tokenizer.eos_token` to ensure that padded batches can be formed.

A central implementation detail is that different Hugging Face model families store their final normalization layer and output head under different attribute names. To make the analysis code reusable, the script defines `get_final_norm_and_head(model)`, which checks common architectures (GPT-NeoX/Pythia, GPT-2, Llama, and MPT) and returns a pair `(final_norm, lm_head)`. For Pythia specifically, this resolves to `model.gpt_neox.final_layer_norm` and `model.embed_out`. In addition, the script stores a reference to the last transformer block `last_layer = model.gpt_neox.layers[-1].eval()`, because later it will apply this final block to intermediate hidden states in order to compute intermediate-layer logits in a way that matches the model's own last-step computation. Specific operations for other model families are similar and can be found in the corresponding code.

Each batch of text is tokenized using `padding="max_length"`, `truncation=True`, and `max_length=1024`, which produces `input_ids` and an `attention_mask` of shape (`batch_size, seq_len`). The analysis is done at next-token prediction positions, so the script forms targets as `targets = input_ids[:, 1:]` and considers prediction sites aligned to `input_ids[:, :-1]`. To ensure that both the conditioning token and the target token are non-padding, it defines a boolean mask

$$\text{valid\_mask} = (\text{attention\_mask}[:, 1:] > 0) \,\&\, (\text{attention\_mask}[:, :-1] > 0),$$

which has shape (`batch_size, seq_len-1`) and is used to filter out padded positions from every saved statistic.

The forward pass calls the model with `output_hidden_states=True`, `use_cache=False`, and `return_dict=True`. This returns `out.hidden_states`, a tuple of length `num_layers+1` containing tensors of shape (`batch_size, seq_len, hidden_size`); index 0 corresponds to the input embedding output, and index `num_layers` corresponds to the final hidden state before the output head. From these hidden states, the script computes per-layer increments `dh[layer] = hidden_states[layer+1] - hidden_states[layer]` for all layers. These quantities are then used to quantify geometric alignment across depth.

The script accumulates several layerwise statistics across all valid tokens, storing them on CPU to keep GPU memory usage bounded. For each layer index `layer` in $0, \ldots, L$, it saves the hidden-state norm $||\text{h\_layer}||$ at prediction positions, implemented as `hidden_states[layer][:,:-1,:].norm(dim=-1)` and filtered by `valid_mask`.

To evaluate predictive quality as a function of depth, the script computes cross-entropy for each layer's representation using a consistent readout pipeline. For the final layer $\ell-1$, it directly uses `out.logits` from the model forward pass and computes per-token cross-entropy at each position as `CE = -log_p(target)` after applying `log_softmax`. For all earlier layers $0, \ldots, \ell-2$, it first applies the model's final transformer block to that layer's hidden state, using `last_layer(hidden_states[layer], attention_mask=prepared_mask, position_ids=position_ids)[0]`. Here `prepared_mask` is a broadcasted attention mask, constructed so that padded keys contribute a large negative value (`torch.finfo(dtype).min`) in attention logits, and `position_ids` are derived by cumulatively counting non-pad tokens so that positions begin at 0 over real to-

kens. The resulting activations are then passed through the model's final layer norm and output head, `logits = lm_head(final_norm(hidden_state_))`, and the script computes per-token cross-entropy values `CE_layer` that are filtered by `valid_mask` and concatenated across batches. Although the script also computes tokenwise entropy `H_layer`, that portion is commented out and is not saved in the current run.

In addition to predictive loss, the script records geometric rotation diagnostics across depth. For each layer $l$, it computes `Theta[l]`, the angle between consecutive hidden states at that token position, defined as

$$\theta(h_l, h_{l+1}) = \arccos\Big(\text{clip}(\cos(h_l, h_{l+1}), -1, 1)\Big),$$

implemented via `F.cosine_similarity(...).acos()` with clipping for numerical stability. It also computes `Theta_dh[l]`, the angle between consecutive depth increments `dh[l]` and `dh[l+1]`, which measures how the direction of the representation update changes from layer to layer. Finally, it computes `angle_to_end[l]`, the angle between the layer-$l$ hidden state and the final hidden state `h_ℓ`, which quantifies how aligned intermediate representations are with the model's terminal representation. All of these angles are computed at prediction positions `(:,:-1,:)` and stored only for `valid_mask` positions.

After processing all `4800` documents, the script prints the wall-clock time for the full run and then stacks the accumulated CPU vectors into dense tensors. The resulting saved arrays have shapes `(N_tokens, ℓ)` for `cross_entropy`, `theta`, and `angle_to_end`; shape `(N_tokens, ℓ-1)` for `theta_dh`; and shape `(N_tokens, ℓ+1)` for `norms`, where `N_tokens` is the total number of valid (non-pad) next-token prediction positions observed across all batches. These tensors are saved with `torch.save` to the output path under the keys `cross_entropy`, `theta`, `theta_dh`, `norms`, and `angle_to_end`, providing a compact dataset for subsequent statistical analysis of depth-dependent behavior in the model.

### A.2. Main Text Figures

From Section A.1, we can obtain $\theta(h_l, h_{l+1})$ for each token (i.e., `Theta[l]`) and $\theta(\Delta h_l, \Delta h_{l+1})$ for each token (i.e., `Theta_dh[l]`). Panels a and e in Figure 2 simply plot the raw data for the first 2000 tokens in the dataset. In total, we evaluated 2M tokens in the FineWeb dataset (Penedo et al., 2024). $\langle \cdot \rangle_{\mathcal{D}}$ is obtained by averaging the value over all the tokens evaluated. And $\langle \cdot \rangle_{\mathcal{D},l}$ is obtained by averaging the value over all the tokens and middle layers (first and last layers excluded). Those averaged values yield panels c and d in Figure 2.

$\theta(h_l, h_{l+1}), l = 0, ..., \ell - 1$, for one token form an $\ell$-dimensional vector. Across the dataset, we can then have as many $\ell$-dimensional vectors as the number of evaluated tokens. After performing PCA on these vectors with `scipy` and choosing the first two principal components to show the data, we obtain panel b in Figure 2. The early stopping vector is constructed as `np.array([np.pi/2]+[0.45]*6+[0.0]*16+[np.pi/2])` ($\ell = 24$ for Pythia-410m) and the evenly in the middle vector is `np.array([np.pi/2]+[0.45]*22+[np.pi/2])`. We project the two $\ell$-dimensional vectors into the subspace spanned by the first two principal components.

### A.3. Additional Results

We tested more model sizes other than 410m in the Pythia suit (Biderman et al., 2023). The results agree with Figure 2, see Figure 7. We tested more model families, Qwen-2.5 (Yang et al., 2024), and OPT (Zhang et al., 2022). The results also agree with Figure 2, see Figure 8 and Figure 9, respectively.

## B. LLM Scaling Laws

### B.1. Methods

We used the extracted data in Besiroglu et al. (2024) from the Chinchilla scaling law paper (Hoffmann et al., 2022). The data contains model size $N$, dataset size $D$, and loss $L$. We next used the table at the end of Hoffmann et al. (2022) to find the width $m$ and depth $\ell$ for each model size $N$.

We next describe how we fit the scaling exponents and prefactors for depth, width, and dataset size in the empirical scaling law model. The high-level goal is to model the validation loss as a sum of independent power-law contributions from model width, model depth, and dataset size, and to estimate both the exponents and prefactors directly from empirical measurements. The fitting procedure is implemented as a nonlinear regression in log-space using gradient-based optimization, followed by an uncertainty analysis using the Jacobian of the residuals.

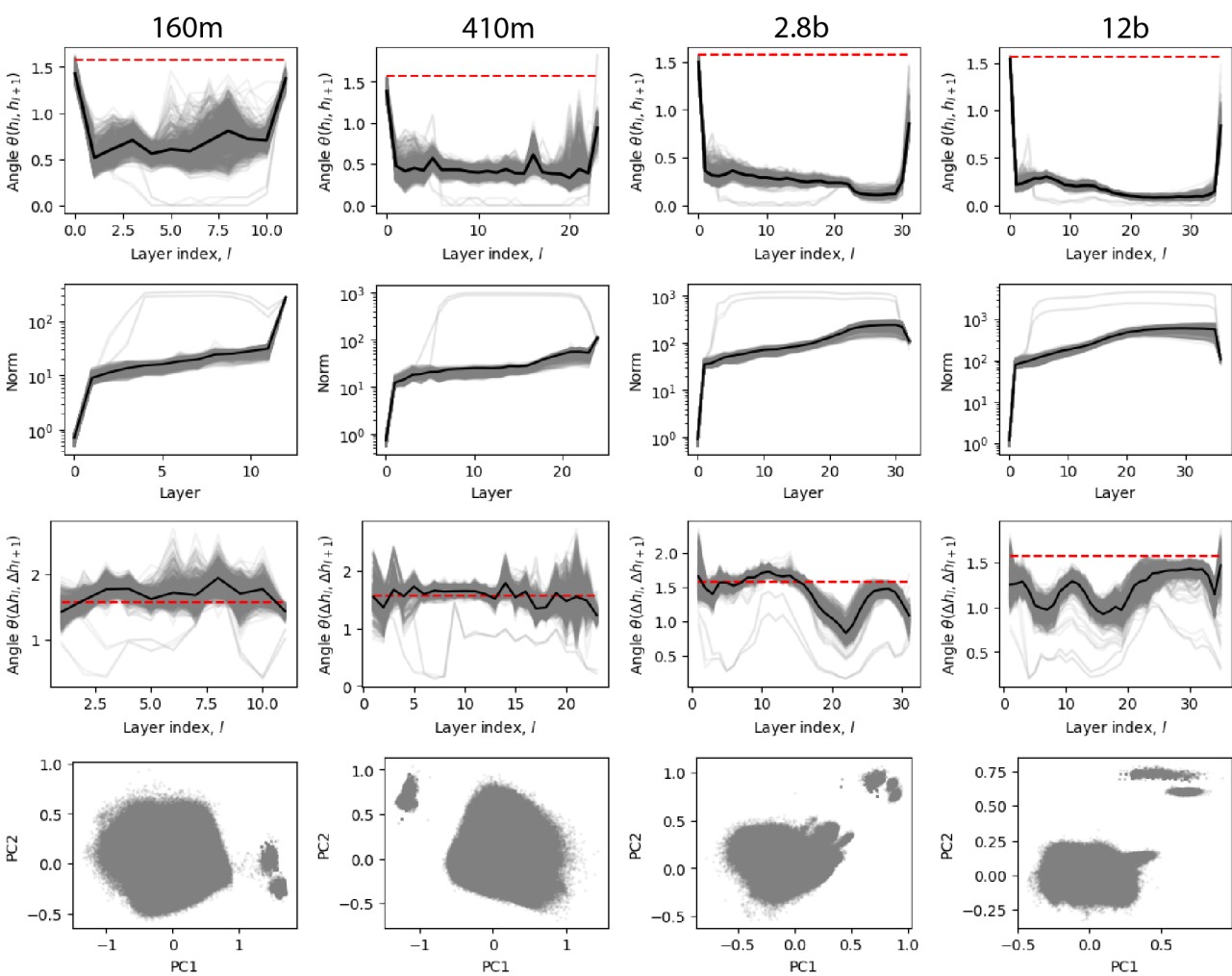

*Figure 7.* Hidden state properties from more model sizes in Pythia. Hidden state norms are included. The results qualitatively agree with Figure 2.

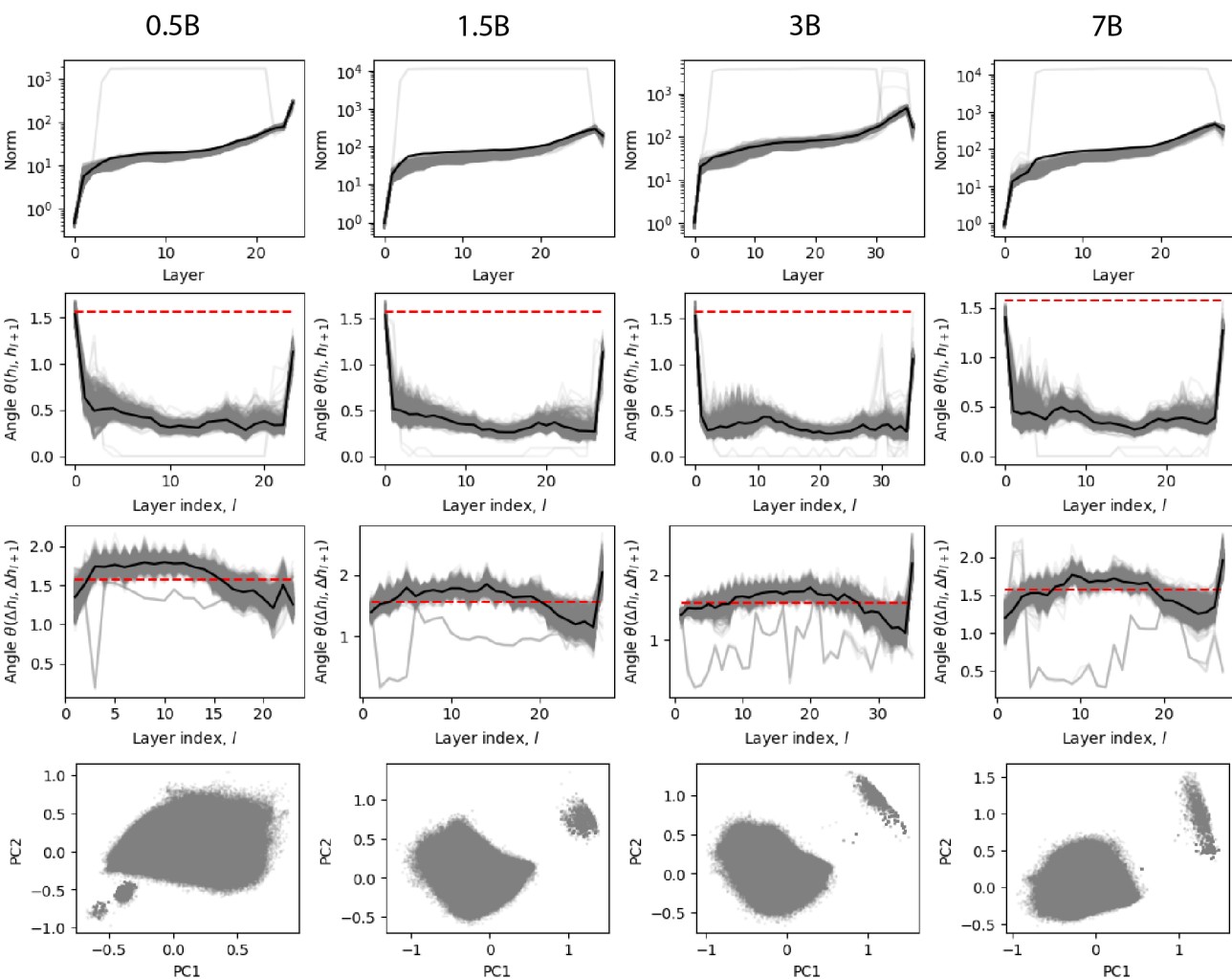

*Figure 8.* Hidden state properties from Qwen-2.5 models (Yang et al., 2024). Hidden state norms are included. The results qualitatively agree with Figure 2.

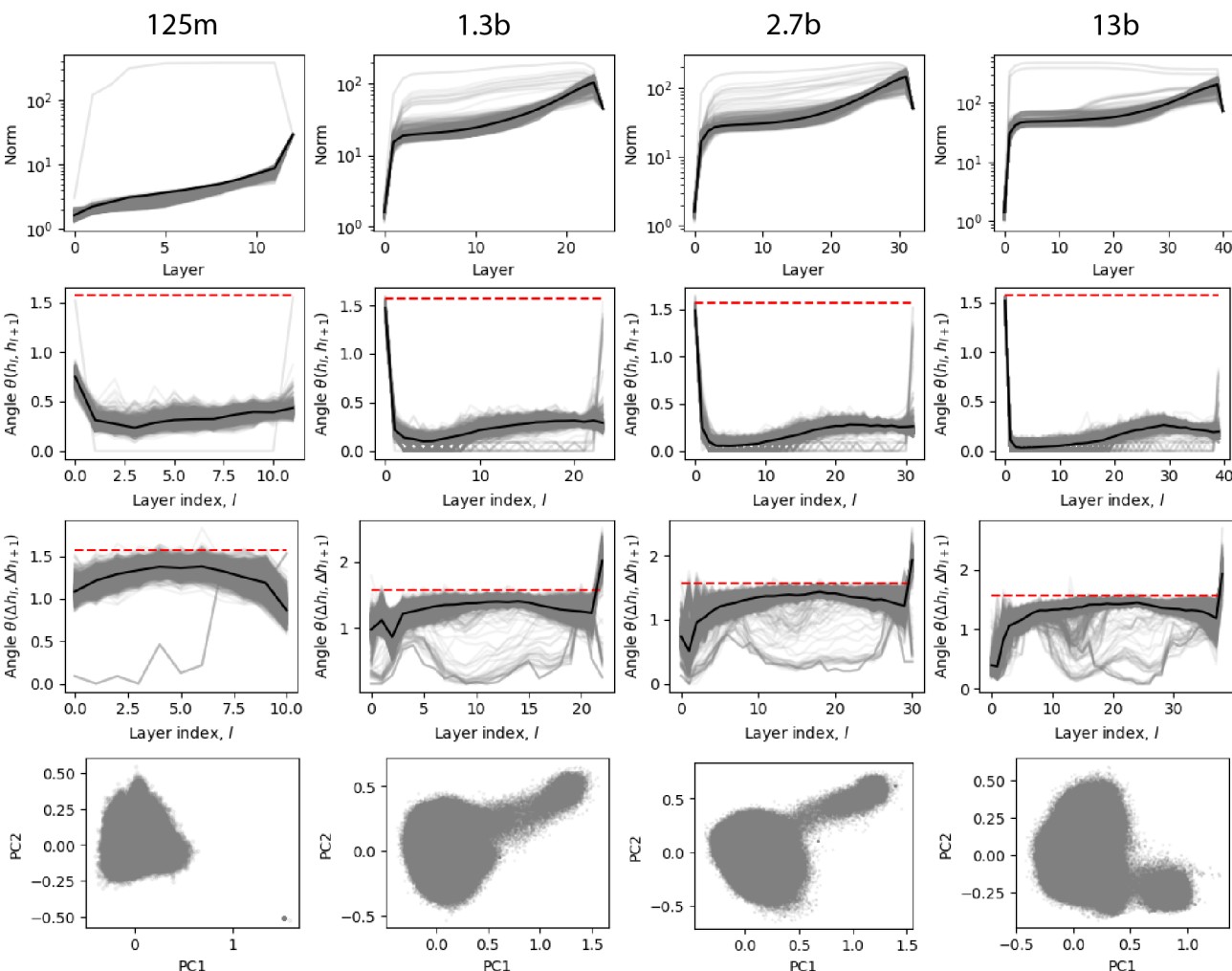

*Figure 9.* Hidden state properties from OPT models (Zhang et al., 2022). Hidden state norms are included. The results qualitatively agree with Figure 2. The OPT models seem to be more different from Pythia and Qwen models. The last layer does not rotate a large angle on average, and the clusters in the PCA plots are very close. We hypothesize that early layers (the first layer excluded) of OPT are in ensemble averaging, and the last few layers, not only the last one, are performing a fixed operation. Although the clusters are less distinguishable, the small one related to the early stopping still occupies a small fraction of tokens, e.g., 1.3% for OPT-13b (we use PC1 $> 0.5$ as the criterion of being in the small cluster).

We load empirical scaling measurements from the file `get_fitting_data_chinchilla.csv`, which contains the model width $m$ (hidden dimension), number of layers $\ell$, dataset size $D$ (number of training tokens), and the observed validation loss. We exclude a subset of high-loss models to avoid outliers dominating the fit. Concretely, we exclude `nr_of_models_excluded = 40` data points with the largest losses, and retain all other data points for fitting. Excluding fewer data points will make the dataset part loss (as is in Figure 11) not like a power law visually after plotting. Excluding more data points will result in similar fitting values but with a larger uncertainty. The scanning of `nr_of_models_excluded` is presented in Section B.3 later. There are 203 data points left. The remaining variables are converted to PyTorch tensors for automatic differentiation.

We assume an additive scaling law of the form

$$L(m, \ell, D) = \frac{c_m}{m^{\alpha_m}} + \frac{c_\ell}{\ell^{\alpha_\ell}} + \frac{c_D}{D^{\alpha_D}} + L_0,$$

where $c_m, c_\ell, c_D$ are positive coefficients, $\alpha_m, \alpha_\ell, \alpha_D$ are scaling exponents, and $L_0$ is an irreducible loss floor. To ensure positivity of the coefficients during optimization, we parametrize them in log-space as $c_x = \exp(\ln c_x)$. The fitted parameters are therefore $(\ln c_m, \ln c_\ell, \ln c_D, \alpha_m, \alpha_\ell, \alpha_D, L_0)$.

We optimize the parameters using the Adam optimizer with separate learning rates for coefficients and exponents:

$$\mathrm{lr}(\ln c_x) = 0.005, \quad \mathrm{lr}(\alpha_x) = 0.0005, \quad \mathrm{lr}(L_0) = 0.005,$$

for a total of `num_epochs = 50000` gradient steps, which ensures convergence. The objective function is the mean squared error in log-loss space,

$$\mathcal{R} = 100 \cdot \mathbb{E}\big[(\ln L_{\mathrm{obs}} - \ln L_{\mathrm{model}})^2\big],$$

which stabilizes optimization across several orders of magnitude in dataset size and model scale.

After convergence, we compute residuals

$$r_i = L_{\mathrm{obs},i} - L_{\mathrm{model},i},$$

and relative residuals $r_i/L_{\mathrm{obs},i}$ for diagnostic plots. To quantify parameter uncertainty, we compute the Jacobian of the residuals with respect to all fitted parameters using PyTorch automatic differentiation. Let $J$ be the Jacobian matrix with entries $J_{i\theta} = \partial r_i/\partial \theta$. Assuming homoscedastic Gaussian noise, the parameter covariance matrix is estimated as

$$\Sigma = \sigma^2 (J^\top J)^{-1},$$

where $\sigma^2$ is the empirical variance of the residuals. Standard errors for each fitted parameter are extracted from the diagonal of $\Sigma$.

Finally, we visualize the fitted scaling law by subtracting the fitted width and depth contributions from the observed losses and plotting the residual dataset-size scaling term on log-log axes.

## B.2. Main Text Figures

The data points in Figure 2f is obtained from

$$L - \frac{c_m}{m^{\alpha_m}} - \frac{c_D}{D^{\alpha_D}} - L_0,$$

where $L$ is actual loss data and other terms are calculated with the fitted values. The line is $\frac{c_\ell}{\ell^{\alpha_\ell}}$ drawn with fitted $c_\ell$ and $\alpha_\ell$.

## B.3. Additional Results

We can do the same to evaluate the width part loss by plotting

$$L - \frac{c_\ell}{\ell^{\alpha_\ell}} - \frac{c_D}{D^{\alpha_D}} - L_0,$$

along with the theory $\frac{c_m}{m^{\alpha_m}}$ (Figure 10).

The dataset part loss is estimated by

$$L - \frac{c_m}{m^{\alpha_m}} - \frac{c_\ell}{\ell^{\alpha_\ell}} - L_0,$$

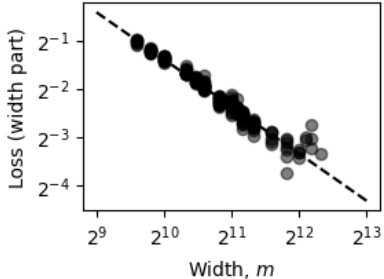

*Figure 10.* Loss due to the width agrees well with a power law. The dashed line is based on $\frac{c_m}{m^{\alpha_m}}$ drawn with fitted $c_m$ and $\alpha_m = 0.98 \pm 0.08$.

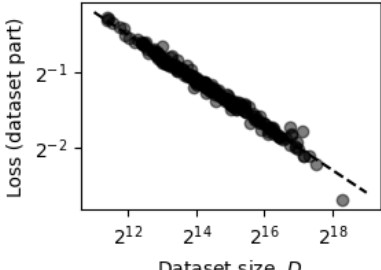

*Figure 11.* Loss due to the dataset size agrees well with a power law. The dashed line is based on $\frac{c_D}{m^{\alpha_D}}$ drawn with fitted $c_D$ and $\alpha_D = 0.30 \pm 0.01$.

which is plotted in Figure 11.

Based on our results that $\alpha_m \approx 1$ and $\alpha_\ell \approx 1$, we should have the optimal width-depth relationship given a fixed number of parameters $N \approx 12m^2\ell$ as $m \propto \ell$ where the coefficient is only related to $c_m$ and $c_\ell$. We next check the relationship between depth and width in actual models (Figure 12). It seems that $\ell \propto m^{0.67}$ can better fit most of the data we have. Yet, when both $m$ and $\ell$ are large enough, it seems the models are following $m \propto \ell$, agreeing with our expectation. We are not sure if previous works (Hoffmann et al., 2022; Brown et al., 2020) scanned depth and width and used the optimal relationship. If they did, there might be two reasons at first $m \propto \ell$ is not satisfied. One is that when $m$ is small contribution from attention layers and language model head to $N$ is significant, whose number of parameters is proportional to $m$ rather than $m^2$, making the dependence of $N$ on $m$ complicated and obscuring the optimal $m$-$\ell$ relation. The second is that when $m$ and $\ell$ are small, some crossing terms of $m$ and $\ell$ may still play a role in loss.

In our observation, the first and the last layers may not be in the ensemble averaging regime. The number of layers in the ensemble averaging regime is better described by $\ell - 2$. And the loss may be a clearer power law with $\ell - 2$. We next try to fit the loss with the form

$$L = \frac{c_m}{m^{\alpha_m}} + \frac{c_\ell}{(\ell - 2)^{\alpha_\ell}} + \frac{c_D}{D^{\alpha_D}} + L_0.$$

It turns out that the newly fitted $\alpha_\ell = 1.1 \pm 0.2$. Compared to the previous one, $\alpha_\ell = 1.2 \pm 0.3$, we indeed obtain a better power law as the standard deviation decreases, and this better power law has an exponent closer to the prediction 1.0. The other newly fitted exponents, $\alpha_m = 0.96 \pm 0.08$ and $\alpha_D = 0.30 \pm 0.01$, remain similar to the old ones. The fact that the fitted power law gets closer to the theoretical expectation after making a more accurate estimation of the number of layers that may be in the ensemble averaging regime, based on hidden state behaviors, supports the argument that those layers are in the ensemble averaging regime.

We plot the loss by parts as before in Figure 13 (width part), Figure 14 (depth part), and Figure 15 (dataset part).

Furthermore, if we plot $\ell - 2$ versus $m$ with data from Hoffmann et al. (2022); Brown et al. (2020), the fitted relationship $\ell - 2 \propto m^{0.74}$ (Figure 16) is closer to the asymptotic optimal relationship $\ell - 2 \propto m$ predicted comparing to previous

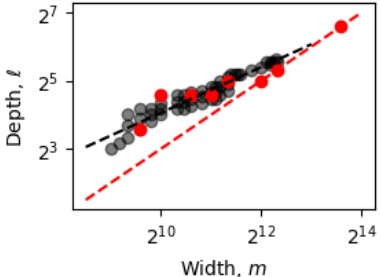

*Figure 12.* Actual width-depth relationship in models. The dark points are from Chinchilla models (Hoffmann et al., 2022) and the red points from GPT-3 (Brown et al., 2020). We find that when both $m$ and $\ell$ are large (the last three red points), $m \propto \ell$ (red dashed line) is true, agreeing with our theory. Chinchilla models are fitted by the dark dashed line, suggesting $\ell \propto m^{0.67}$.

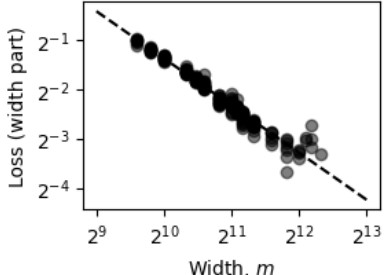

*Figure 13.* Loss due to the width agrees well with a power law (fitted after replacing $\ell$ by $\ell - 2$). The dashed line is based on $\frac{c_m}{m^{\alpha_m}}$ drawn with fitted $c_m$ and $\alpha_m = 0.96 \pm 0.08$.

situation where we studied $\ell$ versus $m$. We should focus on the scaling relationship between the number of layers in the ensemble averaging regime and the width. Though, $\ell - 2 \approx \ell$ asymptotically for large depth $\ell$.

The main text results and the results above ignored 40 data points with the highest losses. The number 40 is obtained by scanning. We used the same fitting procedure as Section B.1 but varying the number of excluded points. The code is in `scanignore.py`. The fitted exponents and their standard errors are plotted in Figure 17. When we include too many points in fitting, the high-loss points are not on the same power law, which yields high standard errors. If we ignore too many points, including those already in the power-law region, the standard errors of fitting will increase as there are fewer points in the fitting. The number 40 is when the standard error of $\alpha_\ell$ started to grow after reaching its lowest value (Figure 17 left). We chose 40 such that we are confident that we ignored all high-loss points not on the power law. Increasing the number of ignored data points further, the fitted exponents remain similar (Figure 17 right). We can conclude that $\alpha_\ell \approx 1$, $\alpha_\ell \approx 1$, and $\alpha_D \approx 0.3$ are robust. If we plot the ignored 40 points onto Figure 2f, Figure 10, and Figure 11, which is shown in Figure 18, we can see that most of the ignored data are mostly at the early training stage and not on the same line as the rest. By choosing 40 as the number of ignored points in fitting, we reported our most confident and accurate results in the main text.

To test the identifiability and robustness of our fitting, we also did a held-out validation by randomly splitting the 203 data used in the main text fitting into 80% training and 20% testing. The code is in `held_out_validation.py`. We tested 10 random splits. The relative training error is $0.41\% \pm 0.02\%$ ($0.02\%$ is the standard deviation across 10 splits), and the relative test error is $0.40\% \pm 0.07\%$. We also added bootstrap stability checks for the 7-parameter fit on the same 203 data. The code is in `bootstrap.py`. We resampled the data with replacement for 200 runs, and the $\alpha_m$ has a mean of 1.00 and a standard deviation of 0.04, the $\alpha_\ell$ has a mean of 1.01 and a standard deviation of 0.02, and the $\alpha_D$ has a mean of 0.30 and a std of 0.02.

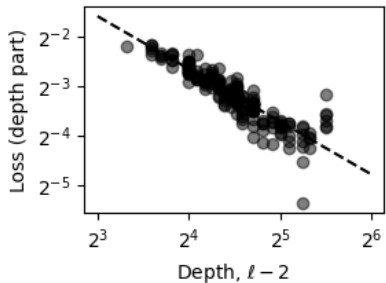

*Figure 14.* Loss due to the depth agrees well with a power law (fitted after replacing $\ell$ by $\ell - 2$). The dashed line is based on $\frac{c_\ell}{(\ell-2)^{\alpha_\ell}}$ drawn with fitted $c_\ell$ and $\alpha_\ell = 1.1 \pm 0.2$.

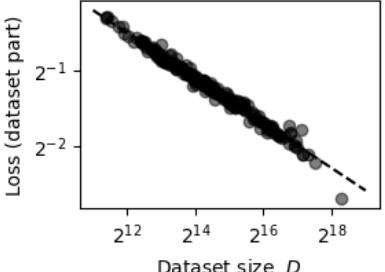

*Figure 15.* Loss due to the dataset size agrees well with a power law (fitted after replacing $\ell$ by $\ell - 2$). The dashed line is based on $\frac{c_D}{m^{\alpha_D}}$ drawn with fitted $c_D$ and $\alpha_D = 0.30 \pm 0.01$.

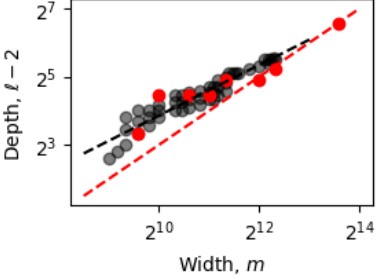

*Figure 16.* Actual width-depth relationship in models. The dark points are from Chinchilla models (Hoffmann et al., 2022) and the red points from GPT-3 (Brown et al., 2020). We find that when both $m$ and $\ell - 2$ are large (the last three red points), $m \propto \ell - 2$ (red dashed line) is true, agreeing with our theory. Chinchilla models are fitted by the dark dashed line, suggesting $\ell - 2 \propto m^{0.74}$.

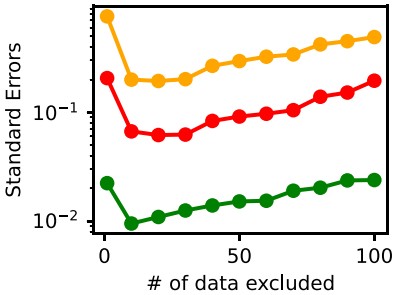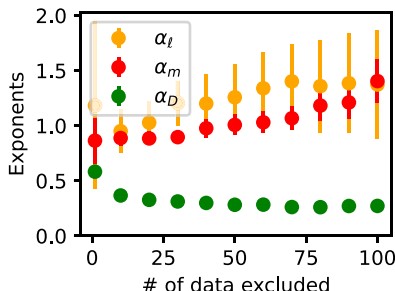

*Figure 17.* Fitted exponents (right) and their standard errors (left) as a function of the number of points with the highest loss ignored in fitting.

"x" represents the 40 points/models excluded in fitting

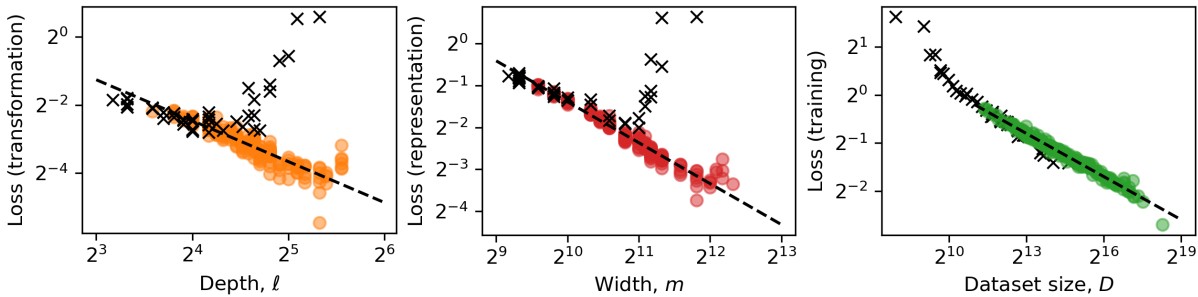

*Figure 18.* Adding the ignored 40 points onto Figure 2f, Figure 10, and Figure 11, respectively.

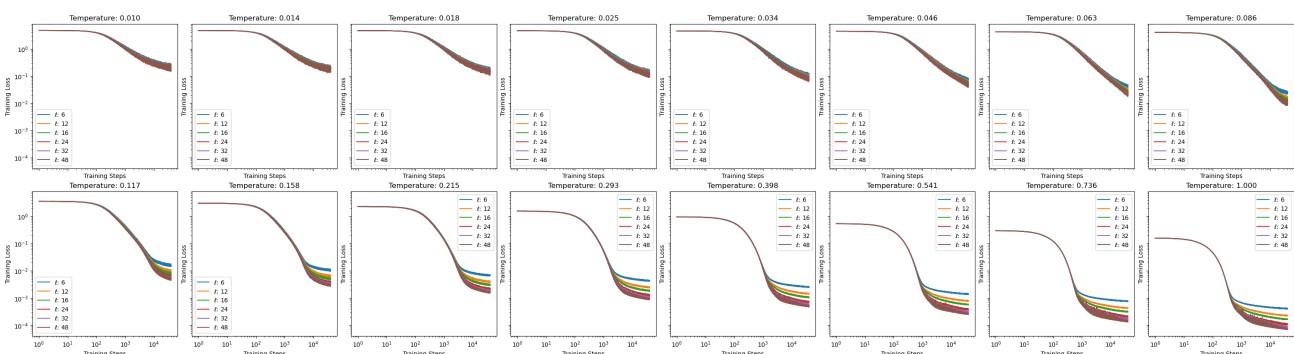

*Figure 19.* Training dynamics of `exp-9.py` for Figure 3. Under high temperatures, the training converge. Under low temperatures, the loss is dominated by training part, not model size part or not limited by depth.

# C. Toy Model Experiments

## C.1. Methods

This experiment studies how learning dynamics depend on the entropy of the target distribution by using a synthetic teacher-student setup. The high-level goal is to generate a controllable next-token distribution from a teacher model with adjustable temperature, train a student model to match the teacher using KL divergence, and measure how training and test loss depend on the teacher entropy. This isolates the effect of peaked versus flat target distributions in a minimal setting, independent of natural language structure.

We run the experiment as a distributed job array. The teacher distribution is controlled by a temperature parameter $T$, with `n_temperatures = 16` logarithmically spaced values between `T_min = 10^{-2}` and `T_max = 1`. For each temperature, we run `n_teachers = 3` replicates of teachers. Evaluation is performed on `n_eval_batches` fresh batches of Gaussian inputs. All models are trained in full precision (FP32) on CPU, and no weight decay or dropout is used.

The model architecture is described in the main text (Section 3).

## C.2. Main Text Figures

Figure 3b is based on code `exp-9.py` (independent teacher weights $\rho = 0$) and `exp-9-1.py` (independent teacher weights $\rho = 1$). The learning rates are fixed as `6e-4`. Other basic information is described in Section C.1. After obtaining the data, we fit the final test loss with Equation (8) via `scipy`. The training dynamics of models $\ell = 6$ with different teacher temperatures and $\rho = 1$ are shown in Figure 4c.

Realizing the difficulty of training in low temperature, we tried to increase the number of training steps to 80000 in Figure 4, a and b, whose code is `exp-9-3.py`. To make training easier, we also gave the student the teacher head and only trained the MLP layers. It turns out that lower temperatures require orders of magnitude longer time to train and cannot converge within our compute budget. We then fit the exponent $\alpha_\ell$ as Equation (8) at different steps and temperatures.

At the end, to ensure convergence and then study hidden states, we use the MSE between the last hidden states to train the models. The fitted exponents agree with theory well (Figure 23). The code is in `exp-9-6.py`. We then study the angle between hidden states and the angle between updates, the same as LLM experiments. Some of the results form Figure 5.

## C.3. Additional Results

The training dynamics, loss versus time, behind Figure 3 is in Figure 19 ($\rho = 0$) and Figure 20 ($\rho = 1$). Under high temperatures, it is obvious that the loss has plateau at the end. While for low temperatures, loss continue to drop and difference due to depth is not obvious, suggesting no convergence. The training dynamics, loss versus time, behind Figure 4, a and b, is in Figure 21. The training dynamics, loss versus time, behind Figure 5 is in Figure 22. We use MSE for Figure 5. The loss scaling of cross-entropy and MSE is the same, i.e., quadratic in the hidden state difference. The measured exponents $\alpha_\ell$ agree with our theory (Figure 23).

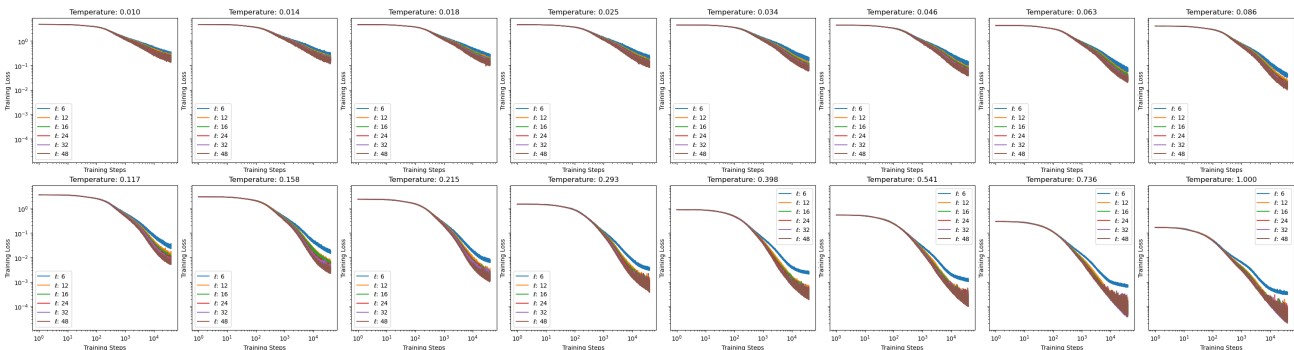

*Figure 20.* Training dynamics of `exp-9-1.py` for Figure 3 and Figure 4c. Under high temperatures, the training converge. Under low temperatures, the loss is dominated by training part, not model size part or not limited by depth.

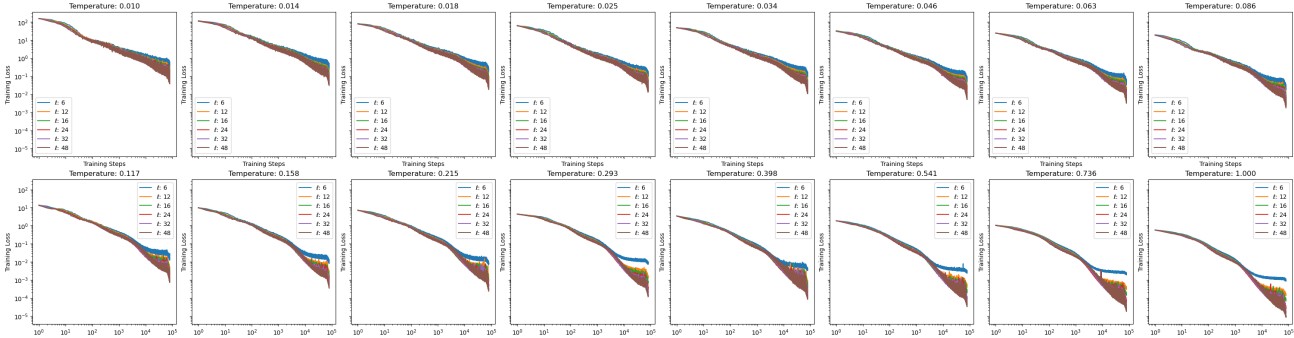

*Figure 21.* Training dynamics of `exp-9-3.py` for Figure 3, a and b. Under high temperatures, the training converge. Under low temperatures, the loss is dominated by training part, not model size part or not limited by depth.

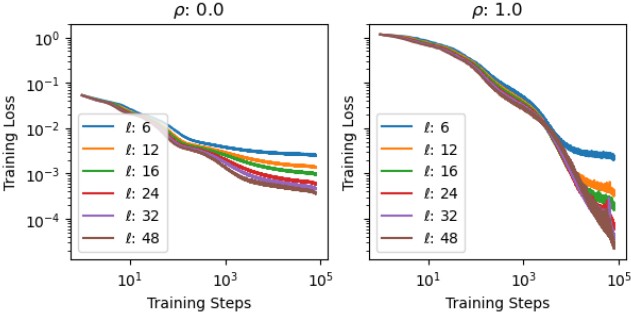

*Figure 22.* Training dynamics of `exp-9-6.py` for Figure 5. We use MSE, which tend to ensure training convergence.

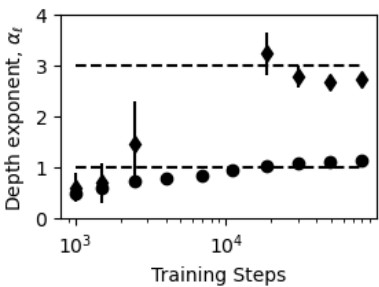

*Figure 23.* Fitted $\alpha_\ell$ from `exp-9-6.py` agree with theory. The diamonds correspond to $\rho = 1$ and dots correspond to $\rho = 0$.

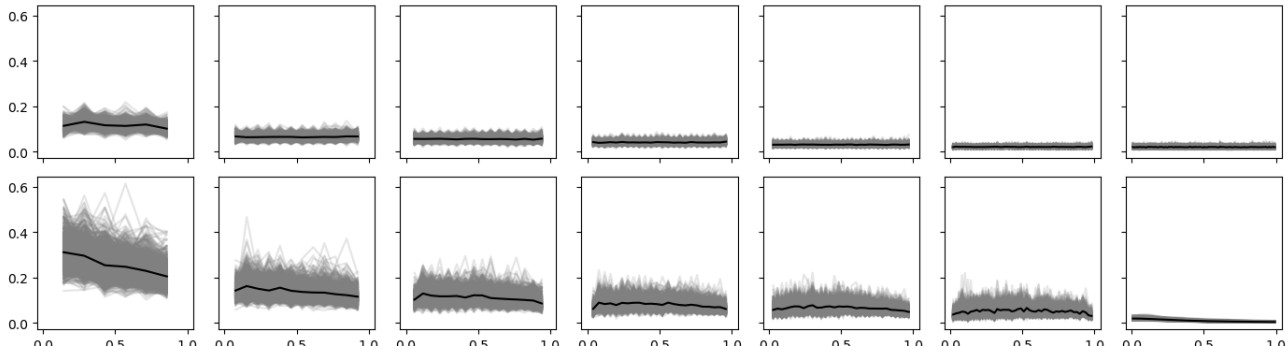

*Figure 24.* Evaluation of hidden state angle $\theta(h_l, h_{l+1})$ after training from `exp-9-6.py`. The first row are from $\rho = 0$ and second row from $\rho = 1$. The columns are from left to right are $\ell = 6, 12, 16, 24, 32, 48$ and the teacher.

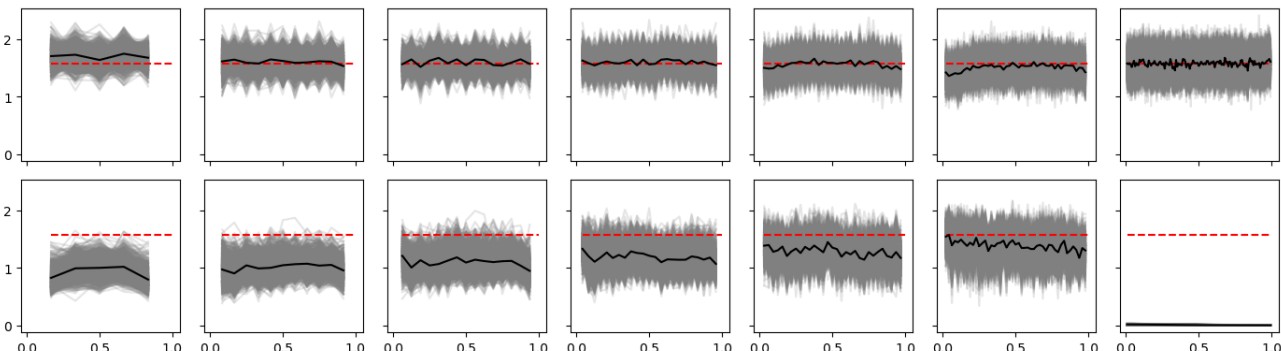

*Figure 25.* Evaluation of hidden state angle $\theta(\Delta h_l, \Delta h_{l+1})$ after training from `exp-9-6.py`. The first row are from $\rho = 0$ and second row from $\rho = 1$. The columns are from left to right are $\ell = 6, 12, 16, 24, 32, 48$ and the teacher.

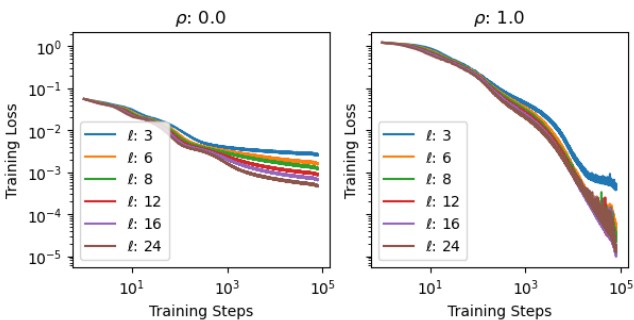

*Figure 26.* Training dynamics of `exp-9-4.py` for Figure 5. We use MSE, which tend to ensure training convergence. The new architecture is used.

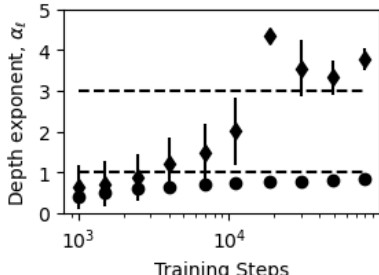

*Figure 27.* Fitted $\alpha_\ell$ from `exp-9-4.py` agree with theory. The diamonds correspond to $\rho = 1$ and dots correspond to $\rho = 0$. The final $\alpha_\ell$ under $\rho = 1$ is not 5, which may be due to the fact larger models have not really converge (Figure 26).

In addition to the hidden states properties shown in Figure 5, we plot all the observed data here from `exp-9-6.py` in Figure 24 (about $\theta(h_l, h_{l+1})$) and Figure 25 (about $\theta(\Delta h_l, \Delta h_{l+1})$). We find that correlation between updates $\theta(\Delta h_l, \Delta h_{l+1})$ is indeed lower under $\rho = 1$ than those under $\rho = 0$, suggesting smooth dynamics under $\rho = 1$ as expected. However, the angles under $\rho = 1$ is still far away from zero, which may be due to limited training or symmetries in the network.

We also did experiments with modified student architecture. The hidden state then propagates through $\ell$ blocks,

$$h_l = h_{l-1} + \text{BLOCK}_l(h_{l-1}), \quad l = 1, \ldots, \ell. \tag{13}$$

Within each block, there are two MLPs,

$$h_{l-1}^c \equiv \frac{1}{2}\text{MLP}_{l,1}(h_{l-1}) + h_{l-1} \tag{14}$$

and

$$\text{BLOCK}_l(h_{l-1}) = \text{MLP}_{l,2}(h_{l-1}^c). \tag{15}$$

Ideally, $h_{l-1}^c$ is to approximate the hidden state at center point $s = (l-1)/\ell + 1/2\ell$ between $(l-1)/\ell$ and $l/\ell$. The two MLPs are approximating the hidden state gradient with $s$. This is a second-order method. If the underlying dynamics is smooth, the discretization error in one layer is $O(\Delta s^3)$. After error accumulation, the upper bound is $O(\Delta s^2)$ and the typical behavior is $O(\Delta s^{5/2})$. The loss based on the upper bound of hidden state error is $L \sim 1/\ell^4$, and the typical behavior is $L \sim 1/\ell^5$. However, if under the ensemble averaging, the theory predicts $L \sim 1/\ell$ regardless of the architecture in one layer. We find that procedural assembly (teacher $\rho = 1$) with this new architecture does have a higher $\alpha_\ell \approx 4$ than that of residual network (see Figure 23), while ensemble averaging (teacher $\rho = 0$) remains $\alpha_\ell \approx 1$ (Figure 27). We therefore further verified that teacher $\rho = 1$ leads to procedural assembly, where a better discretization scheme can help depth scaling, while teacher $\rho = 0$ leads to ensemble averaging, where a better discretization scheme cannot help.

## D. LLM Causal Trace

This appendix provides a self-contained description of the causal tracing experiment in Figure 6 and motivates why the results bear on the functional-group picture. The central question is whether the layer groups visible in causal tracing, within

which many layers are individually capable of recovering the correct prediction, grow proportionally with total model depth, as the ensemble averaging hypothesis predicts. The code is in `causal_trace.py`.

We apply causal tracing (Meng et al., 2022a) to six Pythia models spanning a wide range of depths: 70M (6 layers), 160M (12 layers), 1B (16 layers), 1.4B (24 layers), 2.8B (32 layers), and 12B (36 layers). All models share the same GPT-NeoX architecture, ensuring that differences in tracing results reflect depth alone rather than architectural heterogeneity. The probe sentence is *"The space needle is in downtown Seattle,"* with subject *"space needle."* We write $\ell \in \{1, \ldots, L\}$ for layer index, $i \in \{1, \ldots, T\}$ for token position, and $h_{\ell,i}$ for the hidden state at layer $\ell$ and position $i$. The target output is the token *" Seattle,"* predicted at the position of the last token of *"downtown,"* and $p(\text{Seattle})$ denotes the probability assigned to this token by the softmax over output logits at that position.

The procedure follows three steps. In the clean run, we perform a standard forward pass on the unmodified input, record all hidden states $h_{\ell,i}$, and note the resulting probability $p_{\text{clean}}$, which serves as the performance ceiling. In the corrupted run, we suppress the model's factual recall by adding fixed Gaussian noise to the embedding of every subject token. Concretely, let $\sigma$ denote the standard deviation of the embedding matrix entries; for each subject token at position $i \in [i_s, i_e)$, we add a noise vector $\varepsilon_i \sim \mathcal{N}(0, (3\sigma)^2 I)$ drawn once before all experiments and held fixed thereafter. The corrupted probability $p_{\text{corrupt}}$ is then recorded. Fixing the noise realization is intentional: it ensures the corrupted baseline is identical across every patching experiment, so that any change in the recovered probability is attributable solely to the patch. In the patch runs, for each layer $\ell$ we run a single batched forward pass of $T$ examples, all starting from the corrupted embeddings. In example $j$, we additionally restore the hidden state at position $j$ after layer $\ell$ to its clean value, $h_{\ell,j} \leftarrow h_{\ell,j}^{\text{clean}}$, while all other positions remain corrupted. We record $p_{\ell,j}(\text{Seattle})$ for each pair $(\ell, j)$, and collect the results into a matrix $R \in \mathbb{R}^{L \times T}$ with $R_{\ell,j} = p_{\ell,j}(\text{Seattle})$. Batching over token positions means each layer requires exactly one forward pass, so the full experiment costs $L$ forward passes per model.

We ran the experiment on all six models; however, only Pythia-12B (36 layers) produced a meaningful result, because the smaller models assign negligible probability to *" Seattle"* even in the clean run, rendering the causal tracing signal uninformative. The resulting heatmap for Pythia-12B is shown in Figure 6. Two functional groups of layers emerge. In the earlier group, restoring the hidden state at any of the three subject token positions is sufficient to recover $p(\text{Seattle})$ substantially; this group retrieves factual associations tied to *"The Space Needle"* by integrating information across the subject span. In the later group, restoring the hidden state at the *"downtown"* token position drives recovery; this group extracts *"Seattle"* conditioned on the surrounding geographic context. Within each group, $p_{\ell,j}(\text{Seattle})$ fluctuates across layers without a clear monotone trend, indicating that the relevant knowledge is distributed across multiple layers rather than localized at a single one.

The within-group fluctuation is the key observation. If each layer performed a unique, irreplaceable computation, restoring one of the hidden states after a certain layer would produce the same outcome, as the information is there. Instead, the broad within-group pattern suggests redundant computation: multiple layers store and apply similar information but in a noisy way, and recovering any one is approximately sufficient. This is exactly the signature expected under ensemble averaging, where each layer within a group contributes a noisy, redundant estimate of the same quantity, and their collective output is more reliable than any individual layer's.

The two-group structure has been observed in other large language models. Meng et al. (2022a) report causal tracing results for GPT-2-XL (48 layers; their Figure 1a), GPT-J-6B (28 layers; their Figure 8d), and GPT-NeoX-20B (44 layers; their Figure 8a). Comparing these results with our Pythia-12B (36 layers), the layer span of each functional group scales roughly proportionally with total model depth, which supports the conclusion that depth scaling in this regime primarily increases the size of existing functional groups rather than adding qualitatively new processing stages.

