# OpenReview forum: "Inverse Depth Scaling From Most Layers Being Similar"
_ICML.cc/2026/Conference — ICML 2026 regular_

### Official Review · Reviewer_n63Q · 2026-03-06

**Soundness:** 3
**Presentation:** 3
**Significance:** 2
**Originality:** 3
**Overall Recommendation:** 5
**Confidence:** 2

**Summary:**

The paper observes an inverse scaling with depth. The claim is backed up by a set of real-world experiments, a toy model and a minimal theory. Authors conclude that LLM depth scaling behaviour is dominated by ensemble averaging. This scaling indicates that the network might not optimally benefit from depth. The paper subsequently proposes a method for scaling width and depth.

**Compliance With Llm Reviewing Policy:**

Affirmed.

**Final Justification:**

I thank the authors for their thoughtful and thorough response. I have increased my score, as my concerns have been addressed; however, I still have some uncertainty about my assessment, as reflected in my confidence score.

**Key Questions For Authors:**

I have a few questions to clarify some of the concerns highlighted above and better understand the paper's limitations and contributions.

1. Could you further motivate your choice of toy model?
2. How would different initialisations affect the scaling with depth here?
3. Where does implicit bias come from? Do you have any intuition for why this happened ( initialisation, architecture, optimiser, data) in LLM and not other networks ? Does it happen in a transformer in general? The authors mention 'architectural bias of the residual connection'. Have you tried to study the scaling in small models without the residual connection?
4. The scaling in practice is not exactly the same as the one proposed. Have you tried to train a small model with this scaling?

I look forward to your answer and remain open to increasing my score.

**Limitations:**

The limitations are discussed by the authors.

**Strengths And Weaknesses:**

I would like to first thank the author for their effort and for this interesting work.

**Soundness**

**Strengths:** The paper is technically sound, and the claims are well supported by experimental analysis, while acknowledging the limitations of the theory and toy model.

**Presentation**

**Strengths:** The paper is written in clear English and has a good structure. The figures help the understanding, and are clear.

**Significance**

**Strengths:** The paper points to an interesting phenomenon. The paper is interesting, timely, and highly relevant to the ICML community. Scaling laws are a very relevant bridges between practice and theory.  The paper successfully makes the links between experiments in LLM, theory and toy models, bringing different levels of understanding.

**Weaknesses:** Considering that the scaling proposed is already close to being applied in practice (depth already scales with the width), the paper does not provide a solution to the problem (how could the depth be used more efficiently in transformers?).

The paper does a good literature review, but I point to some literature below that I believe could provide more context. Several papers study the impact of depth on the learning regime ( ie, Saxe et al. 2014, Kunin et al. 2025, Domine et al. 2025, Woodworth et al. 2021). This line of work examines how the learning regime is affected by depth (they do not study LLMs specifically, but could help understand why things are different in LLMs).
Another important line of work has looked into framing the scaling laws as a function of the norm of the parameters instead of their number ( depths or widths) (Wang et al 2025).

---

> ### Author Rebuttal · Authors · 2026-03-30
>
> We thank the reviewer for their insightful comments and suggestions. We will address each of the points in detail below.
>
> **"...the paper does not provide a solution to the problem..."** Our main purpose for this paper is to understand depth scaling. Based on the understanding, we hypothesize that architectures that better utilize compositionality may be more efficient. We may need to add some shortcut connections to encourage early exit. Some existing literature (last paragraph of Sec. 6) also suggests that recurrent use of layers may be beneficial.
>
> **"I point to some literature below that I believe could provide more context. "** Thank you very much! We will read and discuss the references. Some papers mentioned did not study residual connections specifically, and we will discuss this difference and its possible consequences. The current Related Works section is too brief due to the page limit; we will rebalance the sections to include more discussions.
>
> **Q1** Sure. After Fig. 2, we narrowed down to procedural and averaging regimes, which intuitively are closely related to residual connections. We want a toy model to explore these two regimes such that we can distinguish them. We keep the key ingredients, residual connections and cross-entropy loss, but simplify other details like embedding and attention for efficient training.
>
> **Q2** Thanks for the question! Right now, we initialize our weights to be small, similar to the actual LLM training, to facilitate feature learning. In the very early stage of this work, we did initialize the weights to be large, and the depth scaling seemed to be slower. But we believe this may be due to insufficient training. For a finite-size model, once trained to the optimal, we guess the depth scaling will be the same. The scope of our paper is to understand the depth scaling at the optimal.
>
> **Q3** This is a great question we wished to discuss more, but are also afraid to speculate too much. We think the ensemble averaging is not special to LLMs. In fact, it was first proposed in the context of image tasks (Veit et al., 2016). We believe that ensemble averaging shows up in LLMs mainly due to the residual connection and the fact that language transition (from a context to the next possible word) cannot be a smooth dynamics. The residual connection allows the ensemble and the data property prohibits the procedural assembly. If one uses transformers to learn smooth dynamic systems, the transition function may be smooth, and a different depth scaling may show up. We are happy to test this later. We have not studied depth scaling without residual connection as this is a classic topic (adding layers avoids the curse of dimensionality) and there are many works on this. Our understanding from the literature is that without residual connections, the depth scaling is exponential.
>
> **Q4** Thank you for asking! The scaling laws used in practice can be derived from the one we proposed. Given that model size $N \approx 12m^2 \ell$, under the optimal shape, we have $m \propto \ell \propto N^{1/3}$, and the width and depth parts of loss scaling as $1/N^{1 / 3}$. The measured $\alpha_N$ is 0.34 in Chinchilla, very close to be $1/3$. We explored the neural scaling in more detail and explained the mechanisms behind. Yet, our scaling laws are equivalent to the empirical ones near the optimal shape, and can be used in practice. We have not trained any LLM from scratch.

---

> > ### Author Rebuttal · Reviewer_n63Q · 2026-04-02
> >
> > I thank the authors for their thoughtful and thorough response. I have increased my score, as my concerns have been addressed; however, I still have some uncertainty about my assessment, as reflected in my confidence score.

---

> > > ### Author Response · Authors · 2026-04-06
> > >
> > > Thank you very much! We are happy to discuss any follow-up questions if any, which may help to lower your uncertainty.

---

### Official Review · Reviewer_AF7A · 2026-03-13

**Soundness:** 3
**Presentation:** 3
**Significance:** 3
**Originality:** 3
**Overall Recommendation:** 5
**Confidence:** 3

**Summary:**

Scaling laws e.g. Chinchilla have been great rough guides for LLMs and other large models, but generally treat the model size as being described by a simple number, when in practice we often have many different measures of model structure (width, depth, conv filter count...) or other factors such as use of particular activations, residual connections, etc. This paper focuses on depth in particular and finds that depth leads to implicit ensemble formation which cuts error. In general, their results "fill out" gaps in understanding from the simple Chinchilla-level scaling law and try to answer more fundamental questions about model architecture, a very valid angle as the key NN revolution has been around making deeper networks practical on various levels (gradient propagation, residual connections, etc.)

**Compliance With Llm Reviewing Policy:**

Affirmed.

**Final Justification:**

It did not change my evaluation but it was already a strong one.

**Key Questions For Authors:**

1. Prior to the large models era of today, model architecture was iterated on extensively for other ends. How much do your conclusions of a roughly balanced width : depth ratio carry over to past explorations in this matter (e.g. for image classification)

2. Have any closed-source lab architectural details, partially revealed or otherwise, confirmed any part of your hypothesis ? Most leading models are closed-weight.

**Limitations:**

yes

**Strengths And Weaknesses:**

Strengths : The paper studies an important topic, which has industrial significance due to the fact that scaling laws are often used to ballpark estimates of token usage / pre-training setups etc. on a rough level, and it is also written in a very nice "Physics-like" way - assumptions are not made in a contrived manner to prove theorems but rather toy cases are studied well for empirical insight that lines up with intuition.

Weaknesses : The only major LLM used in the paper is Qwen 2.5 (Pythia is too small). This class of LLM has had multiple strange behaviors reported esp with RL rewards which makes the conclusion effectively off of 1 large model by itself.

---

> ### Author Rebuttal · Authors · 2026-03-30
>
> We thank the reviewer for their great summary and inspiring questions. We will address each of the points in detail below.
>
> **Weaknesses** The largest Pythia model we tested is 12B, and the largest Qwen 2.5 has 72B parameters, which are similarly large. We now also added experiments on OPT models (Fig2 in [link](https://anonymous.4open.science/r/depth-rebuttal-1C8C)). The arguments on the hidden state are the same. Qwen models are indeed different from other model families in my experience. Yet, from the perspective of depth scaling and observables we measured, there is no significant difference between Qwen and other families.
>
> **Q1** Thanks for the question! The ensemble averaging of layers picture was proposed in (Veit et al., 2016) cited, which is verified for image tasks. We are not aware of loss scaling working in image tasks, as people usually care about accuracy instead of loss. We believe inverse depth scaling may also hold for image tasks. However, we are not sure if the inverse width scaling is true for image tasks, and therefore not sure whether a constant aspect ratio is optimal for image tasks. The work we cited (Liu et al., 2025a) attributes the inverse width scaling to superposition, which may be specific to LLMs.
>
> **Q2** With the fitted loss formula Eq. (3), we can predict that the optimal width-depth ratio is around 74. And the ratio ranging from 30 to 180 at most increases the reducible compute-optimal loss by 5%. This agrees with the original Kaplan et al. (2020) paper, which shows that the loss is robust to the width-depth ratio near the optimal. Closed-source models usually do not report any details. What we can find is that GPT-3 models reported the widths and depths, and the width-depth ratio ranges from 64 to 128, which is around our predicted optimal ratio, supporting our results. We will add this discussion to the paper.

---

> > ### Author Rebuttal · Reviewer_AF7A · 2026-04-03
> >
> > Thank you for the rebuttal. I don't have further questions. In a dream world, we would also add experiments from larger models as it is well known that empirical work in LLMs has a nasty habit of not generalizing to truly large-scale scenarios (e.g. the well known case of "better-than-Adam" optimizers). But I feel the eval is enough for now. Since 5 is already a good score, I will be keeping it.

---

> > > ### Author Response · Authors · 2026-04-06
> > >
> > > Thank you very much for the suggestions and your support. From reading the technical reports on GPT-4, DeepSeek, etc., our feeling is that the neural scaling laws persist to very large scales. As one component, we tend to believe the depth scaling will also be generalizable. Otherwise, the optimal-compute frontier will change at large scales. From our theory, the ensemble averaging also has no reason to fail on a larger scale. We would like to evaluate larger models if we have more resources.

---

### Official Review · Reviewer_oyNc · 2026-03-13

**Soundness:** 3
**Presentation:** 3
**Significance:** 3
**Originality:** 2
**Overall Recommendation:** 4
**Confidence:** 3

**Summary:**

This paper studies how loss depends on depth in large language models and asks whether depth is used compositionally, as a discretization of smooth dynamics, or as an ensemble of similar layers. The method combines hidden-state diagnostics in pretrained LLMs, an additive scaling-law fit L=c_m/m^{\alpha_m}+c_\ell/\ell^{\alpha_\ell}+c_D/D^{\alpha_D}+L_0, and teacher-student residual toy models designed to separate procedural assembly from ensemble averaging. The main reported findings are that most middle layers update hidden states incrementally rather than showing early stopping, the fitted depth exponent is approximately 1.2\pm0.3, and toy models with non-smooth teacher dynamics reproduce both inverse depth scaling and hidden-state signatures that resemble the LLM observations.

**Compliance With Llm Reviewing Policy:**

Affirmed.

**Key Questions For Authors:**

1. **How robust is the inverse-depth conclusion to the fitting choices in Eq. (3)?** In particular, can you report how \(\alpha_\ell\) changes when the 40 excluded high-loss points in Sec. B.1 are included, and when Eq. (3) is compared directly against alternative depth forms such as exponential/logarithmic decay or models with explicit width-depth interaction terms (Eq. 3; Fig. 2f; Sec. B.1)? A response showing that \(\alpha_\ell \approx 1\) is stable across these choices would substantially increase my confidence in the paper’s central empirical claim; a response showing strong sensitivity would make me view Result 2 as more tentative.

2. **Can you provide a more direct calibration of the “ensemble averaging” interpretation on real LLMs?** Right now, the paper itself notes that the neighboring-update correlation evidence in Fig. 2e is “not definitive” without a calibrated reference, and the strongest mechanistic evidence comes from the toy models (Fig. 2e; Figs. 3-5; Secs. 2-4). If you can map the LLM measurements quantitatively onto the toy-model regimes, or provide a real-LLM intervention/ablation that distinguishes averaging from generic non-smooth dynamics, that would materially strengthen Result 3; otherwise I would interpret the ensemble explanation as plausible but not uniquely established.

3. **Could you clarify the assumptions behind the theoretical transition from Eqs. (10) to (11) and (12)?** In particular, what precise notion of independence is required for the diffusive accumulation argument, and is there any empirical evidence from the toy models that these independence assumptions are approximately satisfied (Eq. 10-12; Sec. 4)? A rigorous clarification, or even an empirical covariance analysis, would increase my confidence in the theoretical interpretation of \(\alpha_\ell \approx 1\); if these steps are mainly heuristic, I think the paper should frame the theory more explicitly as an intuition rather than a derivation.

**Limitations:**

Partially. The paper does a good job acknowledging several **technical** limitations in Sec. 6, including that Eq. (3) is not derived from first principles, that alternative mechanisms could reproduce similar signatures, and that the ensemble structure may be more complicated than a single homogeneous group (Sec. 6). However, the **societal impact** discussion is not yet adequate: the Impact Statement simply says that no specific consequences need to be highlighted (p. 9), which feels too brief for a paper making claims that could influence future LLM architecture and scaling decisions.

I would suggest expanding this section to discuss:

 (i) the risk of overgeneralizing the proposed depth law beyond the studied model families, datasets, and training regimes; (ii) the practical implications for compute allocation, energy use, and efficiency if practitioners act on the width-depth guidance prematurely; and (iii) the fact that the work appears low-risk in terms of direct harmful capabilities, but still has indirect consequences for how the community invests training resources and interprets “efficient” model design.

**Strengths And Weaknesses:**

## Strengths

- **Clear framing of the scientific question**
  - The introduction identifies a concrete gap between monolithic parameter-count scaling laws and depth-specific behavior, which gives the work a crisp motivating question from the outset.
  - The three candidate regimes—compositional assembly, procedural assembly, and ensemble averaging—are defined explicitly before experiments begin, which improves interpretability of later evidence.

- **Real-LLM analysis is substantial and informative**
  - The hidden-state analysis uses 4800 documents and about 2M tokens, which is a meaningful empirical scale for geometric probing and supports stable dataset averages.
  - Figure 2b gives a concrete quantitative split—roughly 99.6% “evenly in the middle” versus 0.4% “early stop”—which directly supports the claim that strong compositional early stopping is not the dominant behavior for most tokens.
  - The paper uses more than one diagnostic on the same phenomenon, combining layer-to-layer hidden-state angles with neighboring-update correlations, which is stronger than relying on a single handcrafted statistic.


- **Effective theory-experiment triangulation**
  - The paper does not stop at empirical plotting; it builds toy teacher-student systems that isolate smooth versus non-smooth target dynamics, which is a sensible way to discriminate between candidate mechanisms.
  - The tied-versus-independent teacher construction operationalizes procedural assembly versus ensemble averaging with a clean control variable, which is an elegant experimental design choice.

## Weaknesses

- **Empirical scaling-law evidence is not yet robust enough**
  - The central quantitative claim depends on the additive decomposition in Eq. (3), yet the paper notes that most available model families do not provide enough width-depth coverage to constrain all terms and therefore falls back to reconstructed data from one family (Eq. 3; Sec. 2; p. 4-5).
  - In the appendix, 40 high-loss points are excluded because otherwise the dataset-size component does not look like a power law visually, and that filtering choice introduces a real risk of selection bias in the fitted exponents (Sec. B.1; p. 14).
  - The reported 0.4% average relative error is encouraging, but No direct evidence found in the paper of held-out validation, cross-validation, or bootstrap stability checks for the seven-parameter fit, so identifiability remains hard to judge (Sec. 2; Sec. B.1).

- **Evidence for ensemble averaging in real LLMs remains indirect**
  - Section 2 explicitly says that the large neighboring-update angles are inconsistent with smooth dynamics but “not definitive” because there is no calibrated reference, so Fig. 2e alone does not settle the mechanism question (Fig. 2e; Sec. 2; p. 4).
  - The jump from toy-model similarity to Result 3 is plausible but indirect, because No direct evidence found in the paper of an intervention on real LLM layers that directly tests variance reduction, averaging behavior, or ensemble-like error cancellation (Secs. 3-4; Fig. 5).

- **Mathematical formulation and notation need tightening**
  - Equation (3) is central to the paper but is introduced as a motivated decomposition rather than a derivation, and the discussion later confirms that it was not obtained from first principles; this weakens the causal interpretation of the fitted \(\alpha_\ell\) (Eq. 3; Sec. 6).
  - Equation (2) hides width-depth interaction terms in an ellipsis and assumes they are subdominant when width and depth are large, but no quantitative sensitivity analysis tests whether that assumption is actually harmless on the fitted dataset (Eq. 2; p. 4).


- **Experimental scope and reproducibility could be stronger**
  - The real-model evidence is most quantitative for the main family in Figure 2, while the additional families are presented mainly as qualitative agreement; reporting family-specific exponents and uncertainty intervals would make the generalization claim sharper (Fig. 2; Figs. 6-7; Sec. A.3).
  - The PCA interpretation in Figure 2b depends on hand-crafted prototype trajectories, including a fixed middle-layer angle of 0.45 rad, which is a useful heuristic but not obviously unique or sensitivity-tested (Sec. A.2; p. 12).

---

> ### Author Rebuttal · Authors · 2026-03-30
>
> We thank the reviewer for their comprehensive summary and constructive suggestions!
>
> **Weakness 1**
> - We now also tested Pythia and OPT models (10 sizes at final checkpoints used for 5-parameter fitting). The fitted $\alpha_m = 1.2\pm 0.3$ and $\alpha_\ell = 1.3 \pm 0.4$. We believe it shows that our results are not specific to Chinchilla.
> - We plotted the fitted exponents versus the number of ignored high-loss points, which quantitatively shows that 40 yields one of the best fits. We also added the ignored points to the loss decomposition plots, where the high-loss points are mostly due to insufficient training. We need to use 40 instead of 30 to make all points on one line. Please see Fig. 1 at the [link](https://anonymous.4open.science/r/depth-rebuttal-1C8C).
> - We added held-out validation by randomly splitting the data into 80% training and 20% testing. We tested 10 random splits. The relative training error is 0.41% $\pm$ 0.02% (0.02% is std across 10 splits), and the relative test error is 0.40% $\pm$ 0.07%. We also added bootstrap stability checks for the 7-parameter fit. We resampled the data with replacement for 200 runs, and the $\alpha_m$ has a mean of 1.00 and a std of 0.04, and the $\alpha_\ell$ has a mean of 1.01 and a std of 0.02, and the $\alpha_D$ has a mean of 0.30 and a std of 0.02.
>
> **Weakness 2**
> - We used hidden state behaviors to reject the compositional assembly hypothesis and then use the measured depth scaling $\alpha_\ell\approx 1$ and toy model theory/experiment to conclude that the averaging regime is the most likely explanation.
> - We guess that by "evidence remains indirect", you mean that we do not have evidence directly based on the hidden states. We are trying to reproduce Fig. 1 of arxiv:2202.05262 for different model sizes. This figure shows that many layers store the same information. If we can show that the region of layers that store the same information increases proportionally with depth, this could be a direct evidence for the averaging regime.
>
> **Weakness 3**
> - Eq. (3) is semi-empirical, where each term may have some theoretical motivation but the additive form is not derived from first principles. This is a problem of all scaling law studies.
> - Yes, this is a good point. We replaced $c_\ell / \ell^{\alpha_\ell}$ with $c / m^{\alpha_m'} \ell^{\alpha_\ell'}$ in Eq. (3) and then did the fitting. The fitting error is low. However, the fitted $\alpha_m'$ and $\alpha_\ell'$ have standard errors close to be themselves, which means the fit is not robust and the form is problematic. We believe this fact shows that the interaction terms are not leading terms or are less important.
>
> **Weakness 4**
> - We added the experiments on Qwen and OPT models (Fig3 and Fig2, respectively in the [link](https://anonymous.4open.science/r/depth-rebuttal-1C8C)). The quantitative results are similar to Pythia.
> - The value is not that important. We want to know what the two clusters are in PCA. Changing 0.45 to 0.2 or 0.6 won't change the fact that the projected point falls into the big cluster.
>
> **Q1** We have partly addressed this in reply to Weaknesses 1 and 3. We did not use exponential/logarithmic decay as it has no theoretical motivation and disagree with the fact that neural scaling laws are power laws. Our formula of width and depth scaling can derive model size scaling $1/N^{1/3}$ near optimal shape given $N \approx 12m^2\ell$, agreeing with Chinchilla scaling where empirical $\alpha_N = 0.34$. This fact further supports our current formula.
>
> **Q2** We partly addressed this in reply to Weakness 2. We think the difficulty of calibrated reference is that we do not know what correlation smooth dynamics will lead to in LLMs. We do have the results in toy models, where the smooth dynamics would have angle between updates near $0.9$, obviously smaller than $\pi /2$. But we do not think we can conclude that this will also be the case for LLMs. The loss scaling is good evidence. Other works like arxiv:2202.05262 support the averaging regime.
>
> **Q3** The assumption is that the difference between teacher and student in one layer is independent from the difference in other layers. In non-smooth dynamics, covariance analysis of differences in different layers supports this assumption as off-diagonal elements are several orders of magnitude smaller. But for smooth dynamics, the analysis does not support the assumption. However, the prediction of loss scaling based on this assumption still holds for the smooth dynamics. One hypothesis we have is that there is some symmetry in the network such that different smooth trajectories can lead to the same final point. The correlation of teacher-student difference in different layers reflects the relationship between these "equivalent" trajectories. We are still trying to understand this phenomenon.
>
> **Limitations** We thank the reviewer for the suggestion. We will add the discussion of social impact following the three dimensions of risks.

---

> > ### Author Rebuttal · Reviewer_oyNc · 2026-04-06
> >
> > Thank you for the detailed rebuttal. It addresses several of my main concerns and increases my confidence in the paper.
> >
> > In particular, the added validation around the scaling-law fit is helpful: the held-out train/test splits, bootstrap stability checks, and additional results on Pythia and OPT all strengthen the central inverse-depth claim. I also appreciate the clarification that the interaction-term variant was tested and found to be much less robust, which is useful context for interpreting Eq. (3). The expanded discussion of the independence assumption behind Eqs. (10)–(12) is also helpful.
> >
> > That said, some concerns remain only partially resolved. The inverse-depth conclusion appears stronger now, but I would still like the final paper to show the robustness results more directly and prominently, especially the sensitivity to excluded high-loss points and the uncertainty of the fitted exponents across model families. On the mechanism side, I still view the ensemble-averaging interpretation for real LLMs as plausible rather than uniquely established: the rebuttal gives a reasonable argument, but the strongest direct evidence still comes from the toy models rather than an LLM-side intervention or calibrated mapping. Finally, the discussion around the theoretical transition from Eqs. (10)–(12) remains somewhat heuristic, and I think the final version should be explicit about which parts are derivation and which parts are interpretation.
> >
> > Overall, the rebuttal meaningfully strengthens the paper, and most of my concerns are reduced. My remaining request is mainly for the final version to make the robustness analyses, mechanistic claims, and theory assumptions more explicit and easier to audit.

---

> > > ### Author Response · Authors · 2026-04-06
> > >
> > > We thank the reviewer for their detailed feedback. We try to address the follow-up questions below.
> > >
> > > **"the sensitivity to excluded high-loss points"**
> > > Thank you, we have addressed this in Fig1 in the [link](https://anonymous.4open.science/r/depth-rebuttal-1C8C), and will add the analysis to the next version of the paper.
> > >
> > > **"the uncertainty of the fitted exponents across model families"**
> > > We appreciate the suggestion. We tested Pythia and OPT models (see our rebuttal) and will add the results to the new version. It is, in general, hard to find abundant public data for fitting. Chinchilla data are one of the most abundant, with ~200 data points.
> > >
> > > **"ensemble-averaging interpretation for real LLMs as plausible rather than uniquely established"**
> > > As promised, we added a mechanistic interpretation type analysis based on hidden states.
> > > - Following arxiv:2202.05262, we input "The space needle is in downtown" into the model, where the correct next token is "Seattle". We record the correct hidden states and then corrupt the embeddings of "The space needle". We then recover the correct hidden state at a given token and layer and see its impact on the output probability of "Seattle" (the color in the figure). It turns out that there are two groups of layers, one has a normalized layer $l/\ell \approx 0.2$, and the other has $l/\ell \approx 0.6$, that are important for recovering the correct output. This phenomenon is consistent across different models (GPT-2XL-48-layers, Pythia-12b-36-layers, and Pythia-1b-16-layers, please see [Fig4 in link](https://anonymous.4open.science/r/depth-rebuttal-1C8C)). The mechanistic interpretation (see details in arxiv:2202.05262) is that the first group of layers is responsible for remembering the connection between "The space needle" and "Seattle", while the second group of layers is responsible for augmenting "Seattle" based on "downtown". One can observe the layer ratio of each group is roughly the same across different model sizes, and the number of layers increases proportionally with depth, which is direct evidence of the averaging picture. We will add this analysis to the new version of the paper. The result from Pythia-1b-16-layers is a bit noisy, as this particular sentence may be difficult for a small model, and the model cannot predict "Seattle" even without corruption. We are trying to do statistics on many inputs.
> > > - It turns out that arxiv:2202.05262 already did analysis on different model sizes over different inputs (see [Fig5 in link](https://anonymous.4open.science/r/depth-rebuttal-1C8C)). Again, this is direct evidence for the averaging picture.
> > > - A lot of other results can also be regarded as evidence for the averaging picture. Arxiv:2505.13898 claimed that a deeper model “spreading out” the same computations over more layers rather than adding qualitatively new late computations. [Anthropic’s cross-layer transcoder work](https://transformer-circuits.pub/2025/attribution-graphs/methods.html) shows that their features explicitly read from one layer and can write to all later layers, and they say this can collapse cases of repeated amplification, where many similar features activate each other across successive layers. Arixv:2409.04185 finds that, in larger underlying models, latents are active at multiple layers more often, consistent with adjacent residual-stream states becoming more similar.
> > >
> > > Given all the evidence, we conclude that the ensemble-averaging is the most likely or the dominant mechanism for the depth scaling, but we agree that it is not uniquely established (see our Result 1). In general, we think LLMs are messy, complex systems, and there may not be a single clean answer. It is not about "black or white", "yes or no", "true or false", but rather "how grey" and "more or less". We tried to make the best conclusion based on the evidence gathered.
> > >
> > > **"the discussion around the theoretical transition from Eqs. (10)–(12) remains somewhat heuristic"**
> > > We are not able to revise the paper during rebuttal, but we will incorporate the suggestions into the new version.
> > >
> > > **"make the robustness analyses, mechanistic claims, and theory assumptions more explicit and easier to audit"**
> > > We thank the reviewer for the summary. Given the answers above, we are curious about whether the reviewer thinks the new evidence is strong enough or not, and whether there are more analyses we should do?

---

### Official Review · Reviewer_fh3D · 2026-03-18

**Soundness:** 2
**Presentation:** 3
**Significance:** 2
**Originality:** 3
**Overall Recommendation:** 3
**Confidence:** 4

**Summary:**

This paper examines how large language models utilize depth and seeks to characterize the quantitative relationship between depth and performance. It provides a theoretical account suggesting that depth scaling is primarily governed by an ensemble-averaging effect, in which individual layers function as redundant, noisy estimators, and increasing the number of layers improves performance by reducing variance through averaging.

**Compliance With Llm Reviewing Policy:**

Affirmed.

**Key Questions For Authors:**

Q1. In Section 2, the paper argues that due to the use of LayerNorm, angular updates of hidden states capture the essential dynamics of representation changes. Could the authors provide a more detailed explanation of why LayerNorm leads to this property?

Q2. The paper analyzes token trajectories by performing PCA on hidden representations across layers (forming a D×L trajectory per token, where D is the hidden dimension and L is the number of layers). Could the authors elaborate on why this analysis is meaningful? In particular, how should we interpret the claim that 99.6% of tokens rotate around approximately 0.45 radians across the middle layers? Do all tokens rotate nearly 0.45 radians?

Q3. Can larger embedding rotations be interpreted as beneficial for model performance? Do the authors have any conjecture for why the first and last layers appear to exhibit larger rotations compared to the middle layers?

Q4. Although the toy example is well designed, incorporating multi-head attention could potentially lead to different depth-scaling behavior, as repeated attention operations are known to reduce the effective rank of embeddings with depth [1]. Could the authors provide a high-level explanation of how introducing multi-head attention into the toy setting might affect the observed dynamics?

[1] Attention is Not All You Need: Pure Attention Loses Rank Doubly Exponentially with Depth (Dong et al., 2021)

**Limitations:**

yes

**Strengths And Weaknesses:**

# Strengths
The paper is clearly written and well organized, making it easy to follow. The authors provide explicit justification for each experimental design choice, which helps ground the study in a more principled framework. The toy-model setup is highly controlled, allowing the authors to isolate and study specific behaviors of deep networks. The research question is also well motivated: since model depth is a critical component of modern architectures, understanding the mechanisms underlying depth scaling is an important problem for the large-scale pre-training community.

# Weaknesses
One concern is that the main conclusion—that no single theoretical regime fully explains LLM behavior and that multiple regimes may coexist—makes the overall message feel somewhat incomplete and provides limited intuition about the mechanisms at play. It would be helpful if the authors could offer stronger hypotheses or interpretations of the observed phenomena.
Additionally, the paper may overlook certain scenarios when distinguishing between procedural assembly and ensemble averaging. In particular, analyzing only angular changes between successive layers may not fully capture the dynamics of representation updates. It may also be important to examine the L2 distance between hl​ and hl−1​. There are cases where the angular difference remains relatively large while the update magnitude becomes progressively smaller across layers. In such situations, the dynamics may appear smoother (procedural assembly), with representations contracting and gradually converging, which could affect how one interprets the underlying mechanism.

---

> ### Author Rebuttal · Authors · 2026-03-30
>
> We thank the reviewer for their thoughtful suggestions and inspiring questions. We will address each of the points in detail below.
>
> **"One concern is that the main conclusion..."** Our full picture is that there are roughly three function groups. The first group of layers mixes information of different tokens, the second group mainly performs memory retrieval, and the third group grabs information from the previous context based on similarity. When increasing the number of layers, each group has more layers, but the relative proportion of each group is roughly the same. Averaging happens in each group, which is the main reason for the loss decrease. We can discuss it and provide more intuitions with results from the literature in Sec. 6. We are also designing experiments to directly test this picture (We try to reproduce Fig. 1 of arxiv:2202.05262 for different model sizes for our revised version).
>
> **"analyzing only angular changes between successive layers may not fully capture the dynamics..."** Thank you! The reviewer's concern is very reasonable. But our results in Sec. A3 can rule out this possibility. We showed that the norms of hidden states are increasing with layers. So, a constant angle means the L2 norm of updates are slowly increasing. We can emphasize this point in the main text.
>
> **Q1** LayerNorm will normalize the norm of the hidden states. For example, RMSNorm will normalize the hidden states to be on a sphere. Because models have LayerNorm before LM head, only the direction of hidden states matters for the final prediction or output. We therefore focus on the angular information. Nevertheless, we reported the hidden state norms in Sec. A3, and the conclusions drawn from the angular updates can actually also be drawn from the norm analysis. After all, the hidden states are high-dimensional and we need some reduction to summarize the dynamics.
>
> **Q2** Thanks for the question. This analysis is meaningful because compositional assembly should have different trajectories from other regimes (In compositional assembly, simpler inputs should require fewer layers: different token hidden states would stop changing at different layers, depending on the input complexity). We therefore used this analysis to reject the compositional assembly because the majority (99.6%) of hidden states keep updating throughout the layers. We interpret the result as 99.6% of the tokens are in the procedural or averaging regime in the middle layers. The rest 0.4% of tokens are not updating in the middle layers. There is a little bit of compositional assembly (We have two groups, 0.4% and 99.6%.), but it is too weak. Strong compositional assembly should have as many token groups in PCA as layers.
>
> **Q3** This is a great question, we do not really have an answer to. A larger rotation may not lead to better performance but may be more efficient. As mentioned earlier, we imagine there are three function groups of layers, and the first (few) layers and last (few) layers are qualitatively different from the middle layers. The first few layers are more like information mixing, combining different tokens to form meaningful concepts, and the last one is more like grabbing information from previous similar parts (the second half of the induction head).
>
> **Q4** This is a great question that we will further address and discuss in the main text. At the high level, having multi-head attention will introduce induction bias like rank collapse, which may require MLP to fight against. Each layer can then have larger errors as some resources are used to fight against the bias. However, as the layers are sitll in the averaging regime (neighboring layers perform similar transformations), the inverse depth scaling remains the same due to the central limit theorem. The loss power law will have a larger coefficient as each layer has a larger error. We look forward to extending our toy models to have attention. Yet, it is difficult to retain the conceptual simplicity, and this requires more careful design.

---

> > ### Author Rebuttal · Reviewer_fh3D · 2026-04-06
> >
> > The rebuttal was helpful and clarified several points. I still have two follow-up questions:
> >
> > 1) Could the authors clarify which parts of the angular-update argument are directly supported by the current evidence, versus more interpretive?
> > 2) For the proposed layer-wise function-group explanation, what is the strongest direct evidence currently supporting this view?

---

> > > ### Author Response · Authors · 2026-04-06
> > >
> > > We thank the reviewer for the further questions. We are happy to discuss them and improve our paper. Our answers are as follows:
> > >
> > > 1. We do not have direct empirical evidence for our angular-update argument. To get direct evidence, we need to design a new experiment: changing the norms of the last hidden states and observing the impact on the outputs. If there is no impact, then we can conclude that direction or angle should be the main focus. In our opinion, we do not really need to do this experiment, as by definition, LayerNorm(x) = LayerNorm(cx) for any positive constant c. So, indeed, we should focus on the direction and, therefore, the update of the direction. If the reviewer thinks that our current storyline is not clear or that this detail is confusing, we are happy to move the norm analysis from the Appendix to the main text. Our conclusion from this analysis is that compositional assembly can be rejected (Result 1), which is still true if we look at the norm updates of hidden states. We are curious about the reviewer's opinions and suggestions.
> > > 2. Thank you for asking. As promised, we added a mechanistic interpretation type analysis based on hidden states, which is the strongest direct evidence we can find now.
> > >     - Following arxiv:2202.05262, we input "The space needle is in downtown" into the model, where the correct next token is "Seattle". We record the correct hidden states and then corrupt the embeddings of "The space needle". We then recover the correct hidden state at a given token and layer and see its impact on the output probability of "Seattle" (the color in the figure). It turns out that there are two groups of layers, one has a normalized layer $l/\ell \approx 0.2$, and the other has $l/\ell \approx 0.6$, which are important for recovering the correct output. This phenomenon is consistent across different models (GPT-2XL-48-layers, Pythia-12b-36-layers, and Pythia-1b-16-layers, please see [Fig4 in link](https://anonymous.4open.science/r/depth-rebuttal-1C8C)). The mechanistic interpretation (see details in arxiv:2202.05262) is that the first group of layers is responsible for remembering the connection between "The space needle" and "Seattle", while the second group of layers is responsible for augmenting "Seattle" based on "downtown". One can observe that the layer ratio of each group is roughly the same across different model sizes, and the number of layers increases proportionally with depth, which is direct evidence of the averaging picture (averaging within each group). We will add this analysis to the new version of the paper. The result from Pythia-1b-16-layers is a bit noisy, as this particular sentence may be difficult for a small model, and the model cannot predict "Seattle" even without corruption. We are trying to do statistics on many inputs.
> > >     - It turns out that arxiv:2202.05262 already did analysis on different model sizes over different inputs (see [Fig5 in link](https://anonymous.4open.science/r/depth-rebuttal-1C8C)). Again, this is direct evidence for the averaging picture, but with (at least) two function groups.
> > >     - A lot of other results can also be regarded as evidence for the averaging picture. Arxiv:2505.13898 claimed that a deeper model “spreading out” the same computations over more layers rather than adding qualitatively new late computations. [Anthropic’s cross-layer transcoder work](https://transformer-circuits.pub/2025/attribution-graphs/methods.html) shows that their features explicitly read from one layer and can write to all later layers, and they say this can collapse cases of repeated amplification, where many similar features activate each other across successive layers. Arixv:2409.04185 finds that, in larger underlying models, latents are active at multiple layers more often, consistent with adjacent residual-stream states becoming more similar.
> > >
> > >     We look forward to hearing the reviewer's opinions on the direct fine-grained Mech. Interp. evidence.

---

### Decision · Program_Chairs · 2026-04-30

**Decision:**

Accept (regular)

**Comment:**

This paper shows that loss scales inversely with depth in LLMs, largely attributed to functionally similar layers acting as an ensemble to reduce variance. The authors support their claims by combining hidden-state analysis on pretrained models with highly controlled toy residual networks.

Reviewers unanimously praised the paper for its clear writing, well-organized structure, and highly relevance of the scientific question, with the experimental methodology highlighted as a major strength. During discussion period, the author provided held-out validation, bootstrap stability checks, and tests on diverse model families which alleviated main methodological concerns. For the camera-ready version, the authors are encouraged to incorporate the robustness analyses and explicitly clarify the boundary between mathematical derivation and heuristic interpretation, as requested by the reviewers. I recommend an acceptance.